# A³: AN ANALYTICAL LOW-RANK APPROXIMATION FRAMEWORK FOR ATTENTION

## ABSTRACT

Large language models have demonstrated remarkable performance; however, their massive parameter counts make deployment highly expensive. Low-rank approximation offers a promising compression solution, yet existing approaches have two main limitations: (1) They focus on minimizing the output error of individual linear layers, without considering the architectural characteristics of Transformers, and (2) they decompose a large weight matrix into two small low-rank matrices. Consequently, these methods often fall short compared to other compression techniques like pruning and quantization, and introduce runtime overhead such as the extra GEMM kernel launches and memory operations for decomposed small matrices. To address these limitations, we propose A³, a post-training low-rank approximation framework. A³ splits a Transformer layer into three functional components, namely QK, OV, and MLP. For each component, A³ provides an analytical solution that reduces the hidden dimension size inside each component while minimizing the component's functional loss (*i.e.*, error in attention scores, attention outputs, and MLP outputs). This approach directly reduces model sizes, KV cache sizes, and FLOPs without introducing any runtime overheads. In addition, it provides a new narrative in advancing the optimization problem from singular linear layer loss optimization toward improved end-to-end performance. Through extensive experiments, we show that A³ maintains superior performance compared to SoTAs. For example, under the same reduction budget in computation and memory, our low-rank approximated LLaMA 3.1-70B achieves a perplexity of 4.69 on WikiText-2, outperforming the previous SoTA's 7.87 by 3.18. We also show versatile applications of A³, including its use in KV cache compression, integration with quantization, fine-tuning and mixed-rank assignments for further performance improvements.

## 1 INTRODUCTION

Large language models (LLMs) have shown exceptional performance in various applications, including language understanding, code completion, and reasoning tasks (Vaswani et al., 2017; Brown et al., 2020; Chen et al., 2021a; Wei et al., 2022). However, these models usually contain billions of parameters, resulting in high computational costs and memory requirements. Linear layers and the attention mechanism contribute significantly to the model size and computational complexity, while the KV cache produced during generation further exacerbates the memory burden.

Low-rank approximation is a promising technique that breaks down a matrix into smaller sub-matrices, directly reducing computational complexity and memory usage without the need of additional specialized hardware support. Usually a trained linear layer $W \in \mathbb{R}^{m \times n}$ is approximated by $\widetilde{W}_r = A_r B_r$, where $A_r \in \mathbb{R}^{m \times r}$ and $B_r \in \mathbb{R}^{r \times n}$ are two rank-$r$ matrices with $r \ll m, n$. At inference time, the original GEMM operation $XW$ is replaced by two smaller GEMM operations $XA_r$ and $(XA_r)B_r$. The challenge is to construct the optimal $A_r$ and $B_r$ that maintains the end-to-end model performance. Recent studies show that minimizing the layer output error instead of the weight error gives better model performance (Zhang et al., 2024a;b; Mozaffari & Dehnavi, 2024), thus various activation-aware methods have been proposed, such as SVD-LLM (Wang et al., 2024),

---

[1]Shared first authorship. Alphabetical by last name.

ASVD (Yuan et al., 2023), FWSVD (Hsu et al., 2022). However, these methods usually target general linear layers, which save FLOPs and memory proportional to $\frac{m+n}{mn}r$ (Note that $\frac{m+n}{mn}r < r$ for any $m, n > 2$). Moreover, these methods rarely consider the architectural characteristics of Transformer, and suffer from severe performance degradation when compared to pruning and quantization.

To address these limitations, we propose $\mathtt{A}^3$, a new analytical framework for post-training low-rank approximation. $\mathtt{A}^3$ splits the Transformer architecture into three functional components: query-key (QK) component, output-value (OV) component, and multi-layer perceptron (MLP) component, and minimizes the functional loss of each component. This three-part decomposition is inspired by the mechanistic interoperability of the Transformer architecture (Elhage et al., 2021), allowing a close alignment of local optimization goals with the overall end-to-end model performance.

We highlight the following contributions of $\mathtt{A}^3$:

- We propose a three-part low-rank approximation setup for multi-head attention (MHA), which formulates the problem as three separate objectives: minimizing the functional loss of (1) QK's attention score, (2) OV's attention output, and (3) MLP's layer output.

- We derive closed-form solutions for the three objectives, which reduces the hidden dimensions shared within each component: QK head dimension, OV head dimension, and MLP intermediate size. This naturally reduces model sizes, KV cache sizes, FLOPs, and avoids runtime overheads like extra GEMM operations. Moreover, $\mathtt{A}^3$ trims both FLOPs/memory and information energy proportionally to the rank $r$, enabling a more effective trade-off between model performance and hardware efficiency.

- We have adapted $\mathtt{A}^3$ for use with diverse Transformer architectures, including group query attention (GQA) and rotary position embedding (RoPE), which allows the application of $\mathtt{A}^3$ across a broad spectrum of models. This overcomes the limitation of existing low-rank approximation methods, which can only be applied on the vanilla MHA architecture.

- We conduct extensive experiments on various LLMs, and show that $\mathtt{A}^3$ outperforms SoTA low-rank methods by a significant margin. For example, our compressed LLaMA 3.1-70B achieves a perplexity of 4.69 on WikiText-2, outperforming SoTA method's 7.87 by 3.18. We also demonstrate further applications of $\mathtt{A}^3$, such as its effectiveness in KV cache reduction, combination with quantization, and mixed-rank assignments for better performance.

## 2 RELATED WORK

**Transformer and its variants** The attention layer and the multi-layer perceptron layer (MLP) are two main modules of the Transformer architecture. In vanilla multi-head attention (MHA) (Vaswani et al., 2017), QK component computes the attention scores as follows:

$$\boldsymbol{A}'_i = \text{row\_softmax}(\boldsymbol{A}_i / \sqrt{d_{\text{qk}}}) = \text{row\_softmax}(\boldsymbol{Q}_i \boldsymbol{K}_i^T / \sqrt{d_{\text{qk}}}) \ . \tag{1}$$

where $\boldsymbol{A}_i$ is the pre-softmax attention score of $i$-th head, and $d_{\text{qk}}$ is the head dimension shared by $\boldsymbol{Q}_i$ and $\boldsymbol{K}_i$. The post-softmax attention score is then multiplied with the value and summed over all heads to form the attention output:

$$\boldsymbol{O} = \sum_i^{h_{\text{q}}} \boldsymbol{A}'_i \boldsymbol{V}_i \boldsymbol{W}_{\text{o},i} \ . \tag{2}$$

Note that there is a head dimension $d_{\text{vo}}$ shared by $\boldsymbol{V}_i$ and $\boldsymbol{W}_{\text{o},i}$. In practice, $\boldsymbol{W}_{\text{o},i}$ are usually concatenated as a single linear layer, $\boldsymbol{W}_o = \left[ \boldsymbol{W}_{o,1}^T, \boldsymbol{W}_{o,2}^T, \dots, \boldsymbol{W}_{o,h_{\text{q}}}^T \right]^T$.

The classic MLP in a Transformer has two linear layers with a ReLU activation function in between:

$$\boldsymbol{X}_{\text{d}} = \text{ReLU}(\boldsymbol{X}_{\text{mlp}} \boldsymbol{W}_{\text{u}}), \quad \boldsymbol{Y}_{\text{mlp}} = \boldsymbol{X}_{\text{d}} \boldsymbol{W}_{\text{d}} \ . \tag{3}$$

$\boldsymbol{W}_{\text{u}}$ and $\boldsymbol{W}_{\text{d}}$ scale the input dimension $d_{\text{m}}$ to the intermediate dimension $d_{\text{inter}}$ and back to $d_{\text{m}}$.

Following the vanilla MHA, numerous Transformer variants have been proposed (Ainslie et al., 2023; Shazeer, 2019; Liu et al., 2024). In recent models, the 2-layer MLP is usually replaced with a 3-layer variant (Shazeer, 2020):

$$\boldsymbol{Y}_{\text{g}} = \boldsymbol{X}_{\text{mlp}} \boldsymbol{W}_{\text{g}}, \quad \boldsymbol{Y}_{\text{u}} = \boldsymbol{X}_{\text{mlp}} \boldsymbol{W}_{\text{u}}, \quad \boldsymbol{X}_{\text{d}} = \text{SiLU}(\boldsymbol{Y}_{\text{g}}) \otimes \boldsymbol{Y}_{\text{u}}, \quad \boldsymbol{Y}_{\text{mlp}} = \boldsymbol{X}_{\text{d}} \boldsymbol{W}_{\text{d}} \ , \tag{4}$$

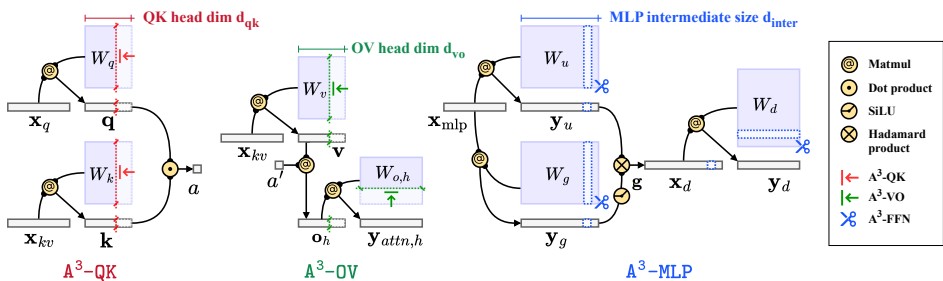

Figure 1: High-level overview of $\mathtt{A}^3$. $\mathtt{A}^3$ performs a low-rank approximation on each $\mathtt{QK}$, $\mathtt{OV}$, and $\mathtt{MLP}$ component, reducing the head dimensions in $\mathtt{QK}$ and $\mathtt{OV}$, and the intermediate dimension in $\mathtt{MLP}$.

where $\otimes$ denotes element-wise multiplication. $\boldsymbol{W}_{\mathrm{u}}$ and $\boldsymbol{W}_{\mathrm{g}}$ upscale the input dimension $d_{\mathrm{m}}$ to $d_{\mathrm{inter}}$ and $\boldsymbol{W}_{\mathrm{d}}$ downscale it back to $d_{\mathrm{m}}$. Group query attention (GQA) (Ainslie et al., 2023) is another widely adopted variant sharing a reduced number of key and value heads among groups of query heads. Besides, many LLMs adopt rotary positional embedding (RoPE) (Su et al., 2024).

Inspired by (Elhage et al., 2021), we split the Transformer layer into three key components: $\mathtt{QK}$ (Equation (1)), $\mathtt{OV}$ (Equation (2)), and $\mathtt{MLP}$ (Equation (3)). We highlight that there are three hidden dimensions, the two head dimensions $d_{\mathrm{qk}}$ and $d_{\mathrm{vo}}$, and one MLP intermediate size $d_{\mathrm{inter}}$, shared within each component but reduced inside the component, while the inter-Transformer layer dimensions are kept at $d_{\mathrm{m}}$. Accordingly, as shown in Figure 1, our method $\mathtt{A}^3$ consists of three parts, $\mathtt{A}^3\text{-}\mathtt{QK}$, $\mathtt{A}^3\text{-}\mathtt{OV}$, and $\mathtt{A}^3\text{-}\mathtt{MLP}$, to compresses the three hidden dimensions.

**Low-rank approximation for compressing Transformers** The linear layer takes a simple form but contributes most parameters to Transformer. For compressing large Transformer models like LLMs, low-rank approximation has been widely studied (Chen et al., 2021b; Saha et al., 2024). Usually a trained linear layer $\boldsymbol{W} \in \mathbb{R}^{m \times n}$ is approximated by $\boldsymbol{W} \approx \widetilde{\boldsymbol{W}}_r = \boldsymbol{A}_r \boldsymbol{B}_r$, where $\boldsymbol{A}_r \in \mathbb{R}^{m \times r}$ and $\boldsymbol{B}_r \in \mathbb{R}^{r \times n}$ are two rank-$r$ matrices that effectively reduce the number of parameters and FLOPs with a small enough $r$. The problem is how to find the optimized $\boldsymbol{A}_r$ and $\boldsymbol{B}_r$ that maintains the model performance. If the objective is to minimize the Frobenius (or spectral) norm of the weight error,

$$\mathrm{argmin}_{\widetilde{\boldsymbol{W}}_r} \|\boldsymbol{W} - \widetilde{\boldsymbol{W}}_r\|_F^2 \quad \text{s.t.} \quad \mathrm{rank}(\widetilde{\boldsymbol{W}}_r) = r \,, \tag{5}$$

according to Eckart-Young theorem (Eckart & Young, 1936a), the optimal solution is to perform truncated singular value decomposition (SVD) on the weight matrix $\boldsymbol{W}$.

$$\boldsymbol{W} = \boldsymbol{U}\boldsymbol{\Sigma}\boldsymbol{V}^T, \quad \widetilde{\boldsymbol{W}}_r = \mathrm{SVD}_r(\boldsymbol{W}) = \boldsymbol{U}_{:,:r}\boldsymbol{\Sigma}_{:k,:r}\boldsymbol{V}_{:r,:}^T \,, \tag{6}$$

where $\boldsymbol{U} \in \mathbb{R}^{m \times k}$ and $\boldsymbol{V} \in \mathbb{R}^{k \times n}$ are the left and right singular vectors and $\boldsymbol{\Sigma} \in \mathbb{R}^{k \times k}$ is the diagonal matrix of singular values. Recent studies show that minimizing the layer output error instead of the weight error gives better end-to-end model performance (Zhang et al., 2024a;b),

$$\mathrm{argmin}_{\widetilde{\boldsymbol{W}}_r} \|\boldsymbol{X}\boldsymbol{W} - \boldsymbol{X}\widetilde{\boldsymbol{W}}_k\|_2^2 \quad \text{s.t.} \quad \mathrm{rank}(\widetilde{\boldsymbol{W}}_k) = r \,, \tag{7}$$

where $\boldsymbol{X} \in \mathbb{R}^{l \times d}$ denotes the activation of $\boldsymbol{W}$. The objective above minimizes the expected $l_2$-norm of layer output error. Recent studies find the optimal solution is:

$$\widetilde{\boldsymbol{W}}_k = (\boldsymbol{R}_{\mathbb{XX}}^{\frac{1}{2}})^{-1}\mathrm{SVD}_r(\boldsymbol{R}_{\mathbb{XX}}^{\frac{1}{2}}\boldsymbol{W}) \,, \tag{8}$$

assuming $\boldsymbol{R}_{\mathbb{XX}}$ is positive definite, where $\boldsymbol{R}_{\mathbb{XX}} = \frac{1}{l}\boldsymbol{X}^T\boldsymbol{X}$ is the autocorrelation matrix with respect to $\boldsymbol{X}$, and $\boldsymbol{R}_{\mathbb{XX}}^{\frac{1}{2}}$ denotes the unique symmetric square root of $\boldsymbol{R}_{\mathbb{XX}}$.

Interestingly, there are several works proposing or leveraging the solution in Equation (8) for various applications. DRONE (Chen et al., 2021b) and SVD-LLM (Wang et al., 2024) directly apply the solution to approximate all linear layers in Transformers. QERA (Zhang et al., 2024b) uses it to build high-precision low-rank terms to compensate for output quantization error, while

CALDERA (Saha et al., 2024) further proposes iterative methods to quantize the low-rank terms, achieving performant sub-2.5-bit post-training quantization. Palu (Chang et al., 2024) reduces KV cache size by decomposing key and value weight matrices with the solution and caching smaller intermediate activations instead of original keys and values. ESPACE (Sakr & Khailany, 2024) extends the base $L_2$ objective by incorporating outlier, gradient information, and per-layer selection of the calibration strategy to maximize fine-tuning accuracy recovery. SLiM (Mozaffari & Dehnavi, 2024) and Oats (Zhang & Papyan, 2024) incorporate sparsity with low-rank to achieve an overall compression gain. Beyond low-rank methods, quantization techniques such as Quarot (Ashkboos et al., 2024), AWQ (Lin et al., 2024), GPTQ (Frantar et al., 2022) and HQQ Badri & Shaji (2023) are also widely used for model compression, and they are often compatible with low-rank approaches for multiplicative gains such as Palu, SLiM and QERA. We show that $\text{A}^3$ can also be effectively combined with quantization @Reviewer gBeN and creates a continuous spectrum of compression levels between discrete quantization points, yielding a way better Pareto frontier at extreme compression.

However, these works target general linear layers and minimize the linear layer output error without considering architectural characteristics. In this work, we step forward to the optimization for functional components. We propose analytical low-rank approximation methods of compressing the QK, OV, and MLP components that minimize the functional errors of attention scores, attention outputs, and MLP outputs, respectively.

## 3 THE $\text{A}^3$ FRAMEWORK

In this section, for each component (QK, OV, MLP), we define the problem (optimization objectives), clarify the assumptions if any, and propose our analytical solutions. The proof for each lemma and theorem is provided in the Appendix. We also provide notation tables in Tables 4 and 5 in the appendix for the ease of reading.

### 3.1 $\text{A}^3$-QK

In the QK component, each head computes its pre-softmax attention scores between queries and keys:

$$\boldsymbol{A}_i = \boldsymbol{Q}_i\boldsymbol{K}_i^T = \boldsymbol{X}\boldsymbol{W}_{\text{q},i}\boldsymbol{W}_{\text{k},i}^T\boldsymbol{X}^T = \boldsymbol{X}\boldsymbol{W}_{\text{qk},i}\boldsymbol{X}^T \; , \tag{9}$$

where $\boldsymbol{X}_{\text{q}} \in \mathbb{R}^{l_{\text{q}} \times d_{\text{m}}}$, $\boldsymbol{X}_{\text{kv}} \in \mathbb{R}^{l_{\text{kv}} \times d_{\text{m}}}$ are the input of query layer and key/value layer respectively, and $\boldsymbol{W}_{\text{qk},i} := \boldsymbol{W}_{\text{q},i}\boldsymbol{W}_{\text{k},i}^T$ denotes the fused weight matrix of the $i$-th head. We seek for the low-rank approximation of $\boldsymbol{W}_{\text{qk},i}$ that minimizes the error of pre-softmax attention scores.

**Problem 1** (Minimization of the pre-softmax attention score error). Given a pretrained Transformer layer, for the $i$-th head of QK component $\boldsymbol{A}_i = \boldsymbol{X}_{\text{q}}\boldsymbol{W}_{\text{qk},i}\boldsymbol{X}_{\text{kv}}^T$ and its rank-$r$ approximated form $\widetilde{\boldsymbol{A}}_i = \boldsymbol{X}_{\text{q}}\widetilde{\boldsymbol{W}}_{\text{qk},i}\boldsymbol{X}_{\text{kv}}^T$, approximating the head by minimizing the error between $\boldsymbol{A}_i$ and $\widetilde{\boldsymbol{A}}_i$ on a calibration set:

$$\operatorname*{argmin}_{\widetilde{\boldsymbol{W}}_{\text{qk},i}} \|\boldsymbol{X}_{\text{q}}(\boldsymbol{W}_{\text{qk},i} - \widetilde{\boldsymbol{W}}_{\text{qk},i})\boldsymbol{X}_{\text{kv}}^T\|_F^2 \quad \text{s.t.} \quad \text{rank}(\widetilde{\boldsymbol{W}}_{\text{qk},i}) = r \; , \tag{10}$$

where $\boldsymbol{X}_{\text{q}} \in \mathbb{R}^{l_{\text{q}} \times d_{\text{m}}}$ and $\boldsymbol{X}_{\text{kv}} \in \mathbb{R}^{l_{\text{kv}} \times d_{\text{m}}}$ denote the inputs for query and key/value projections, which can also be considered as the calibration set when $l_{\text{q}}$ and $l_{\text{kv}}$ are sufficiently large.

**Lemma 1** (Equivalent form of Problem 1). *The objective in Problem 1 is equivalent to:*

$$\operatorname*{argmin}_{\widetilde{\boldsymbol{W}}_{qk,i}} \|\boldsymbol{R}_{\mathbb{X}_q\mathbb{X}_q}^{\frac{1}{2}}(\boldsymbol{W}_{qk,i} - \widetilde{\boldsymbol{W}}_{qk,i})\boldsymbol{R}_{\mathbb{X}_{kv}\mathbb{X}_{kv}}^{\frac{1}{2}}\|_F^2 \; , \tag{11}$$

*where $\boldsymbol{R}_{\mathbb{X}_q\mathbb{X}_q} := \frac{1}{l_q}\boldsymbol{X}_q^T\boldsymbol{X}_q$ and $\boldsymbol{R}_{\mathbb{X}_{kv}\mathbb{X}_{kv}} := \frac{1}{l_{kv}}\boldsymbol{X}_{kv}^T\boldsymbol{X}_{kv}$ are the autocorrelation matrices of the query and key/value calibration activations respectively, and $\boldsymbol{R}_{\mathbb{X}_q\mathbb{X}_q}^{\frac{1}{2}}$, $\boldsymbol{R}_{\mathbb{X}_{kv}\mathbb{X}_{kv}}^{\frac{1}{2}}$ denote the corresponding unique symmetric matrix square roots.*

The complete derivation of Lemma 1 is given in Section B.1.1.

**Theorem 2** ($\text{A}^3$-QK for MHA-NoPE). *The optimal solution to Problem 1 is*

$$\widetilde{\boldsymbol{W}}_{qk,i} = \left(\boldsymbol{R}_{\mathbb{X}_q\mathbb{X}_q}^{\frac{1}{2}}\right)^{-1}\text{SVD}_r\left(\boldsymbol{R}_{\mathbb{X}_q\mathbb{X}_q}^{\frac{1}{2}}\boldsymbol{W}_{qk,i}\boldsymbol{R}_{\mathbb{X}_{kv}\mathbb{X}_{kv}}^{\frac{1}{2}}\right)\left(\boldsymbol{R}_{\mathbb{X}_{kv}\mathbb{X}_{kv}}^{\frac{1}{2}}\right)^{-1} \; , \tag{12}$$

*where $\text{SVD}_r(\cdot)$ denotes the truncated SVD operator.*

The proof of Theorem 2 is given in Appendix B.1.2. In practice, we apply Theorem 2 to all pairs of QK heads ($i = 1, \ldots, h_q$), and assign

$$\widetilde{\boldsymbol{W}}_{q,i} := \left( \boldsymbol{R}_{\mathbb{X}_q \mathbb{X}_q}^{\frac{1}{2}} \right)^{-1} \boldsymbol{U}_{:,:k}, \quad \widetilde{\boldsymbol{W}}_{k,i}^T := \boldsymbol{\Sigma}_{:k,:k} \boldsymbol{V}_{:k,:}^T \left( \boldsymbol{R}_{\mathbb{X}_{kv} \mathbb{X}_{kv}}^{\frac{1}{2}} \right)^{-1} , \tag{13}$$

where $\boldsymbol{U}_{:,:k}$, $\boldsymbol{\Sigma}_{:k,:k}$, and $\boldsymbol{V}_{:k,:}^T$ are the truncated SVD components given by Theorem 2. This gives approximated query and key weights with a new smaller head dimension $r < d_{qk}$. Note that though Theorem 2 is performed on each head separately, the low-rank head weights can still be concatenated together and implemented as a single linear layer.

## 3.2  A$^3$-OV

Expand the summation over all OV head outputs in Equation (2). The matrix form of the attention layer output $\boldsymbol{O} \in \mathbb{R}^{l_q \times d_m}$ can be expressed as

$$\boldsymbol{O} = \sum_{i=1}^{h_q} \boldsymbol{O}_i = \sum_{i=1}^{h_q} \boldsymbol{A}_i' \boldsymbol{X}_{kv} \boldsymbol{W}_{v,i} \boldsymbol{W}_{oi} = \sum_{i=1}^{h_q} \mathbf{P}_i \boldsymbol{W}_{vo,i} , \tag{14}$$

where $\boldsymbol{P}_i := \boldsymbol{A}_i' \boldsymbol{X}_{kv} \in \mathbb{R}^{l_q \times d_m}$ is the product between post-softmax attention score and the input matrix of key/value layer, and $\boldsymbol{W}_{vo,i} := \boldsymbol{W}_{v,i} \boldsymbol{W}_{oi} \in \mathbb{R}^{d_m \times d_m}$ denotes the fused weight matrix of the $i$-th head. Now each term $\boldsymbol{O}_i$ takes the form of a linear layer $\boldsymbol{O}_i = \boldsymbol{P}_i \boldsymbol{W}_{vo,i}$. If $\widetilde{\boldsymbol{O}}_i = \boldsymbol{P}_i \widetilde{\boldsymbol{W}}_{vo,i}$ denotes the approximated $\boldsymbol{O}_i$, the upper bound of the attention output error can be derived as follows:

$$\|\boldsymbol{O} - \widetilde{\boldsymbol{O}}\|_2^2 = \|\sum_{i=1}^{h_q} (\boldsymbol{O}_i - \widetilde{\boldsymbol{O}}_i)\|_2^2 \leq \sum_{i=1}^{h_q} \|\boldsymbol{O}_i - \widetilde{\boldsymbol{O}}_i\|_2^2 . \tag{15}$$

Though $\|\boldsymbol{O} - \widetilde{\boldsymbol{O}}\|_2^2$ can be directly minimized via matrix stacking and truncated SVD, its closed-form solution incurs a higher computational cost, as elaborated in Section B.2.3. Here, we treat minimizing each error term $\|\boldsymbol{O}_i - \widetilde{\boldsymbol{O}}_i\|_2^2$ as an independent problem. Thus the optimal solution to $\widetilde{\boldsymbol{W}}_{vo,i}$ is already given by Equation (8).

**Problem 2** (Minimization of per-head attention output error). Given a pretrained Transformer layer, for the $i$-th head of OV component $\boldsymbol{O}_i = \boldsymbol{P}_i \boldsymbol{W}_{vo,i}$ and its approximated form $\widetilde{\boldsymbol{O}}_i = \boldsymbol{P}_i \widetilde{\boldsymbol{W}}_{vo,i}$, approximating the head by minimizing the head output error is to minimize the following loss:

$$\mathrm{argmin}_{\widetilde{\boldsymbol{W}}_{vo,i}} \|\boldsymbol{P}_i (\boldsymbol{W}_{vo,i} - \widetilde{\boldsymbol{W}}_{vo,i})\|_F^2 \quad \text{s.t.} \quad \mathrm{rank}(\widetilde{\boldsymbol{W}}_{vo,i}) = r . \tag{16}$$

**Theorem 3** (A$^3$-OV for MHA-NoPE). *The optimal solution to Problem 2 is*

$$\widetilde{\boldsymbol{W}}_{vo,i} = \left( \boldsymbol{R}_{\mathbb{X}_{\boldsymbol{P}_i} \mathbb{X}_{\boldsymbol{P}_i}}^{\frac{1}{2}} \right)^{-1} \mathrm{SVD}_r (\boldsymbol{R}_{\mathbb{X}_{\boldsymbol{P}_i} \mathbb{X}_{\boldsymbol{P}_i}}^{\frac{1}{2}} \boldsymbol{W}_{vo,i}) , \tag{17}$$

*where $\boldsymbol{R}_{\mathbb{X}_{\boldsymbol{P}_i} \mathbb{X}_{\boldsymbol{P}_i}} = \frac{1}{l_q} \boldsymbol{P}_i^T \boldsymbol{P}_i$ is the autocorrelation matrix of $\boldsymbol{P}_i$ and $\boldsymbol{R}_{\mathbb{X}_{\boldsymbol{P}_i} \mathbb{X}_{\boldsymbol{P}_i}}^{\frac{1}{2}}$ denotes its unique symmetric matrix square root.*

Similar to QK component, in practice we assign

$$\widetilde{\boldsymbol{W}}_{v,i} := \left( \boldsymbol{R}_{\mathbb{X}_{\boldsymbol{P}_i} \mathbb{X}_{\boldsymbol{P}_i}}^{\frac{1}{2}} \right)^{-1} \boldsymbol{U}_{:,:k}, \quad \widetilde{\boldsymbol{W}}_{vo,i} := \boldsymbol{\Sigma}_{:k,:k} \boldsymbol{V}_{:k,:}^T \left( \boldsymbol{R}_{\mathbb{X}_{\boldsymbol{P}_i} \mathbb{X}_{\boldsymbol{P}_i}}^{\frac{1}{2}} \right)^{-1} ,$$

to get the approximated value and output weights of head-$i$ with a smaller head dimension $r < d_{vo}$.

## 3.3  A$^3$-MLP

The non-linear activation function in MLP component prohibits us from directly applying SVD. Instead, we first derive an objective for minimizing the MLP output error, and uses CUR decomposition (Mahoney & Drineas, 2009) to find the low-rank form of MLP weights.

**Problem 3** (Minimization of `MLP` output error). Given a pretrained down projection layer $Y_{\text{mlp}} = X_d W_d$ in MLP and its approximated low-rank form $\widetilde{Y}_{\text{mlp}} = \widetilde{X}_d \widetilde{W}_d = X_d U W_d$, minimizing the MLP output error is to minimize the following loss:

$$\operatorname{argmin}_{U=\operatorname{diag}(u_1, u_2, \ldots, u_{d_{\text{inter}}})} \| X_d U W_d - X_d W_d \|_F^2 \quad \text{s.t.} \quad \operatorname{rank}(U) = r , \tag{18}$$

where $X_d \in \mathbb{R}^{l_{\text{down}} \times d_{\text{inter}}}$ is the matrix of intermediate activation vectors, and $U \in \mathbb{R}^{d_{\text{inter}} \times d_{\text{inter}}}$ is a diagonal matrix determining which $r$ columns of $W_d$ to keep.

**Lemma 4** (Equivalent form of Problem 3). *The objective in Problem 3 is equivalent to the following error on the calibration dataset:*

$$\operatorname{argmin}_{U=\operatorname{diag}(u_1, u_2, \ldots, u_{d_{\text{inter}}})} \| R_{\mathbb{X}_d \mathbb{X}_d}^{\frac{1}{2}} U W_d - R_{\mathbb{X}_d \mathbb{X}_d}^{\frac{1}{2}} W_d \|_F^2 \quad \text{s.t.} \quad rank(U) = r , \tag{19}$$

*where $R_{\mathbb{X}_d \mathbb{X}_d} := \frac{1}{l_{down}} X_d^T X_d$ is the autocorrelation matrix of the calibration set $X_d$.*

The derivation of Lemma 4 is in Section B.3.1. This CUR approximation is a well-studied NP-hard problem, and various CUR methods have been proposed (Boutsidis & Woodruff, 2014; Drineas et al., 2006). We thus pick a simple but effective solution from Drineas et al. (2006) and name this approach `A`$^3$`-MLP`.

Following Drineas et al. (2006), we build $U$ by sorting the F-norm of the outer product between the coloumns of $R_{\mathbb{X}_d \mathbb{X}_d}^{\frac{1}{2}}$ and the rows of $W_d$:

$$\lambda_i = \| r_i^T w_i \|_F^2 = \| r_i \|_2^2 \cdot \| w_i \|_2^2 , \tag{20}$$

where $r_i$ is the $i$-th column of $R_{\mathbb{X}_d \mathbb{X}_d}^{\frac{1}{2}}$ and $w_i$ is the $i$-th row of $W_d$. Then $U$ is built by selecting the indexes that gives the top-$r$ $\lambda_i$:

$$U = \operatorname{diag}(u_1, u_2, \ldots, u_{d_{\text{inter}}}) \quad \text{where} \quad u_i = \frac{1}{r\lambda_i} \text{ if } i \in \text{top-}r(\lambda_i) \text{ else } 0 . \tag{21}$$

In practice, we compute $\lambda_i$ for all $i = 1, \ldots, d_{\text{inter}}$. Then we select the $r$ rows of $W_d$ that has $r$ largest non-zero $\lambda_i$ to form $\widetilde{W}_d \in \mathbb{R}^{r \times d_m}$. Accordingly, $\widetilde{W}_u, \widetilde{W}_g \in \mathbb{R}^{d_m \times r}$ are formed by selecting the corresponding columns of $W_u$ and $W_g$ respectively.

Note that most related works, *e.g.*, the ones introduced in Section 2, target general linear layers and replace a weight matrix with two low-rank matrices $XW \approx (XA_r)B_r$, which introduces one more GEMM operation per linear layer at inference time. In contrast, *all of our three solutions only reduce the hidden dimensions of the components ($h_q$, $d_{vo}$, and $d_{inter}$), resulting in the same number of GEMM operations with smaller problem sizes. This naturally enables reduced model sizes, saved the FLOPs of both linear layers and attention, compressed KV cache, without introducing any runtime overhead.*

### 3.4 ADAPTING A$^3$ FOR GQA AND ROPE

Our `A`$^3$`-QK` and `A`$^3$`-OV` described above is for standard MHA. In this subsection, we extend `A`$^3$ for GQA and RoPE, enabling a wider range of applicable models.

**Joint SVD for GQA (A$^3$-QK and A$^3$-OV for GQA-NoPE)** In GQA, a key head is shared with multiples query heads in the same `QK` group. This grouping prevents us from applying Theorem 2 to each `QK` head independently. Inspired by (Ji et al., 2025), we first concatenate the scaled error matrices in Equation (12) within the same `QK` group and apply joint SVD:

$$\operatorname{SVD}\left( \begin{bmatrix} R_{\mathbb{X}_q \mathbb{X}_q}^{\frac{1}{2}} W_{\text{qk},1} R_{\mathbb{X}_{kv} \mathbb{X}_{kv}}^{\frac{1}{2}} \\ \vdots \\ R_{\mathbb{X}_q \mathbb{X}_q}^{\frac{1}{2}} W_{\text{qk},g} R_{\mathbb{X}_{kv} \mathbb{X}_{kv}}^{\frac{1}{2}} \end{bmatrix} \right) = U \Sigma V^T , \tag{22}$$

where $g := \lfloor h_q / h_{kv} \rfloor$ is the number of query heads in this group. Then for this group, we assign

$$\widetilde{W}_{\text{qk},i} := R_{\mathbb{X}_q \mathbb{X}_q}^{\frac{1}{2}}{}^{-1} U_{id_m:(i+1)d_m, :r} , \quad \widetilde{W}_{\text{k, shared}} := \Sigma_{:r, :r} V_{:r, :d_m}^T \left( R_{\mathbb{X}_{kv} \mathbb{X}_{kv}}^{\frac{1}{2}} \right)^{-1} , \tag{23}$$

to build the $i$-th head's approximated query weights $\widetilde{W}_{q,i}$ and the shared head key weights $\widetilde{W}_{\text{k,shared}}$. The subscript with colons, *e.g.*, $\Sigma_{:r,:r}$, denotes array slicing. Similarly, we can apply joint SVD to the `OV` component by concatenating the scaled error matrices along the column dimension. A detailed description can be found in Section B.4.

**CUR Approximation for MHA with RoPE (A³-QK for RoPE)**    Recent works have shown that not all RoPE frequencies are helpful for the model performance (Barbero et al., 2024; Ji et al., 2025). These findings motivate our adaptation of A³-QK for attention with RoPE. RoPE inserts a position-dependent operation before the dot product between queries and keys:

$$\text{RoPE}(\boldsymbol{q}_{\text{q},i}, m, \boldsymbol{k}_{\text{k},i}, n) = \boldsymbol{q}_{\text{q},i}\boldsymbol{\Phi}_m\boldsymbol{\Phi}_n^T{\boldsymbol{k}_{\text{k},i}}^T \,, \tag{24}$$

where $m$ and $n$ are the position indexes of $\boldsymbol{q}_{\text{q},i}$ and $\boldsymbol{k}_{\text{k},i}$, $\boldsymbol{\Phi}_m, \boldsymbol{\Phi}_n \in \mathbb{R}^{d_{\text{qk}} \times d_{\text{qk}}}$ are matrices rotating adjacent pairs of query elements and key elements. To deal with these pairwise rotations, we use CUR approximation to solve the problem in Lemma 1. Similar to A³-MLP, we seek for a rank-$r$ CUR approximation of $(\boldsymbol{R}_{\mathbb{X}_q\mathbb{X}_q}^{\frac{1}{2}}\boldsymbol{W}_{\text{q},i})(\boldsymbol{W}_{\text{k},i}^T\boldsymbol{R}_{\mathbb{X}_{kv}\mathbb{X}_{kv}}^{\frac{1}{2}})$ that extracts the most important head dimensions as well as RoPE frequencies. Assign $\boldsymbol{L} := \boldsymbol{R}_{\mathbb{X}_q\mathbb{X}_q}^{\frac{1}{2}}\boldsymbol{W}_{\text{q},i}$ and $\boldsymbol{R} = \boldsymbol{W}_{\text{k},i}^T\boldsymbol{R}_{\mathbb{X}_{kv}\mathbb{X}_{kv}}^{\frac{1}{2}}$, the objective is

$$\text{argmin}_{\boldsymbol{U}=\text{diag}(u_1,u_2,...,u_{d_{\text{qk}}})}\|\boldsymbol{L}\boldsymbol{U}\boldsymbol{R} - \boldsymbol{L}\boldsymbol{R}\|_F^2 \quad \text{s.t.} \quad \text{rank}(\boldsymbol{U}) = r \,, \tag{25}$$

Instead of sorting by the product of $l_2$-norm, *i.e.*, $\lambda_i = \|\boldsymbol{L}_{:,i}\|_2^2 \cdot \|\boldsymbol{R}_{i,:}\|_2^2$, for $i = 0, 1, \ldots, d_{\text{qk}} - 1$, we sort by the sum of $\lambda_i$ of adjacent pairs (Check Section B.5). This will drop pairs of less important columns in $\boldsymbol{L}$ and rows in $\boldsymbol{R}$, as well as the corresponding pairs of RoPE frequencies. Our adaptation for RoPE can be combined with the joint SVD for GQA, allowing A³ to be applied to various models. The evaluation in Section 4 includes standard MHA, MHA with RoPE, and GQA with RoPE.

## 4 EXPERIMENTS

**Baselines**    We compare A³ against a range of baselines, including vanilla low-rank approximation using SVD and weight-magnitude-based column/row pruning, as well as SoTA approaches, including FWSVD (Hsu et al., 2022), ASVD (Yuan et al., 2023), SVD-LLM (Wang et al., 2024), SVD-LLM v2 (Wang et al., 2024), Palu (Chang et al., 2024), Wanda (Sun et al., 2023), and CLOVER (Meng et al., 2025). However, only several baselines support approximating all the three components (QK, OV, MLP), including SVD, FWSVD, ASVD, SVD-LLM, and SVD-LLM-v2. We conduct a comprehensive comparison against these methods. For other baselines, we align the components to approximate and present results in the ablation study. @Reviewer n81d We define the compression ratio based on parameter count to ensure consistent comparison across all baselines. Unless otherwise specified, all subsequent experiments are conducted under this matched-compression setup.

**Models and benchmarks**    Our evaluation covers vanilla Transformer and its variants, including MHA without RoPE, denoted as MHA-NoPE, (MPT (Team et al., 2023)), MHA-RoPE (LLaMA 1&2 (Touvron et al., 2023a;b)), and GQA-RoPE (LLaMA 3.1 (Grattafiori et al., 2024), Phi 3 (Abdin et al., 2024)). We evaluate on pretraining tasks (WikiText-2 (Merity et al., 2016), C4 (Raffel et al., 2020), and SlimPajama (Shen et al., 2023)) using SVD-LLM's perplexity evaluation code snippet, and downstream tasks (ARC-Challenge, BoolQ, Winogrande, GSM8K (strict match), and MMLU) using lm-eval-harness (Gao et al., 2024). All experiments are post-training low-rank approximation *without fine-tuning*. We use 128 random 2048-token sequences from SlimPajama for all evaluations, except in Figure 2, where we calibrate on WikiText2 to match SVD-LLM's setup (see Section C for details).

### 4.1 MAIN RESULTS

This section presents the main evaluation results where we compare A³ against all the baselines that can be applied to all of the three main components (QK, OV, MLP) in Transformer. We first simply evaluate on LLaMA-7B and eliminate less promising baselines, then conduct a comprehensive evaluation on more models and tasks. Lastly, we present profiling results to highlight A³'s improvement on hardware efficiency.

**Preliminary experiments**    We first apply plain SVD, FWSVD, SVD-LLM, and A³ on LLaMA-7B with 20%, 40%, and 60% compression ratios and compare the perplexity (PPL ↓) results on WikiText-2 to find the most promising baselines. As shown in Figure 2, SVD-LLM and A³ achieve perplexities smaller than others by two to three orders of magnitude. We thus conduct further experiments on SVD-LLM and A³.

**Pretraining tasks and downstream tasks**  In Table 1, we include more models to compare $A^3$ against SVD-LLM on WikiText-2, C4, and SlimPajama, covering MHA-RoPE and GQA-RoPE architectures. $A^3$ outperforms SVD-LLM by a large margin most of the time. Remarkably, $A^3$ achieves a perplexity of 4.69 on WikiText-2 with LLaMA 3.1-70B, which is 3.18 lower than SVD-LLM's 7.87 (a perplexity reduction of 58.6%) at 10% compression ratio. We present the downstream task results in Table 2, where $A^3$ consistently outperforms SVD-LLM in terms of average accuracy (↑) across all five tasks. We observe that the advantage of $A^3$ is more pronounced when the compression ratio is small (10%). We attribute this to the adaptation of $A^3$ for RoPE and GQA, which we will discuss in Section 4.2. Results on Phi 3 are presented in Appendix G.

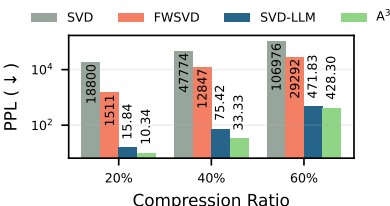

Figure 2: LLaMA-7b PPL on C4, compared to SVD, FWSVD and SVD-LLM.

Table 1: A comparison of perplexity (↓) on WikiText2, C4, and SlimPajama.

| Model | Method | 10% | | | 20% | | |
| --- | --- | --- | --- | --- | --- | --- | --- |
| | | WikiText-2 | C4 | SlimPajama | WikiText-2 | C4 | SlimPajama |
| LLaMA-2-7B | SVD-LLM | 8.78 (+3.30) | 11.73 (+4.14) | 9.49 (+3.35) | 11.58 (+6.1) | 14.91 (+7.32) | 11.93 (+5.79) |
| (MHA-RoPE) | $A^3$ | **5.96 (+0.48)** | **8.34 (+0.74)** | **6.68 (+0.54)** | **7.22 (+1.73)** | **9.91 (+2.31)** | **7.91 (+1.77)** |
| LLaMA-2-13B | SVD-LLM | 7.09 (+2.19) | 9.98 (+2.92) | 7.95 (+2.26) | 9.03 (+4.13) | 12.35 (+5.29) | 9.75 (+4.06) |
| (MHA-RoPE) | $A^3$ | **5.32 (+0.42)** | **7.65 (+0.59)** | **7.65 (+1.97)** | **6.24 (+1.34)** | **8.99 (+1.92)** | **7.15 (+1.47)** |
| LLaMA-3.1-8B | SVD-LLM | 19.12 (+12.86) | 19.37 (+9.33) | 15.14 (+7.57) | 42.28 (+36.02) | 33.6 (+23.56) | 27.44 (+19.86) |
| (GQA-RoPE) | $A^3$ | **7.93 (+1.67)** | **12.56 (+2.52)** | **9.52 (+1.94)** | **11.36 (+5.1)** | **17.87 (+10.29)** | **13.58 (+3.54)** |
| LLaMA-3.1-70B | SVD-LLM | 7.87 (+5.07) | 11.3 (+3.76) | 8.43 (+2.94) | 9.75 (+6.95) | **13.77 (+6.23)** | 10.44 (+4.95) |
| (GQA-RoPE) | $A^3$ | **4.69 (+1.90)** | **8.83 (+1.31)** | **6.59 (+1.10)** | **8.32 (+5.52)** | 13.94 (+6.40) | **10.02 (+4.53)** |

**Higher inference throughput**  The low-rank approximation methods that target general linear layers only save the FLOPs of GEMM in linear layers, but induce runtime overhead like extra GEMM kernel launches and read/write for small matrices. In contrast, $A^3$ saves the GEMM FLOPs in both linear layers and attention, without inducing these overheads. We profile prefilling throughput measured in TPS (tokens/sec) of LLaMA-2-13B on an A100 40GB, and visualize the speedup in Figure 3. SVD-LLM only has speedup for aggressive compression, while $A^3$ always achieves a speedup, higher than SVD-LLM. More runtime analysis can be found in Appendix 11.

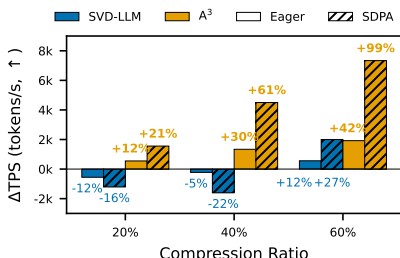

Figure 3: Performance comparisons in Tokens per Second (TPS) of $A^3$ and SVD-LLM (LLaMA-2-13b, A100 40GB, batch size=2, sequence length=2048, attention backend=Eager/SDPA).

## 4.2 ABLATION
## STUDY AND IMPROVEMENT OVER OTHER BASELINES

We conduct ablation studies to evaluate $A^3$'s impact on individual components (QK, OV, MLP). We also include baselines that can be applied to the target components to show $A^3$'s advantage.

**Attention without RoPE**  Theorem 2 ($A^3$-QK) and Theorem 3 ($A^3$-OV) provide optimal solutions for MHA-NoPE's QK and OV components without the need of adaptation. We evaluate the increased perplexity ($\Delta$PPL↓) of $A^3$-QK and $A^3$-OV on MPT-7B (MHA-NoPE) in Figure 4a, comparing against CLOVER (Meng et al., 2025) and Palu (Chang et al., 2024), with compression ratio=20%. CLOVER is equivalent to $A^3$-QK but assumes $R_{X_q X_q}$ and $R_{X_{kv} X_{kv}}$ are identity matrices (no activation information). The bars are grouped by the component being approximated (QK, OV). We add two bars representing simplified version $A^3$-QK in the QK group, $A^3$-Q-only and $A^3$-K-only. $A^3$-Q-only (K-only) replaces the autocorrelation matrix of key (query) in Equation (12) with an identity matrix. We also add a simplified version $A^3$-OV in the OV group, $A^3$-Xkv, which replaces the autocorrelation matrix of $p_i$ in Equation (17) with the autocorrelation matrix of $x_{kv}$. The auto-correlation matrices of these simplified versions of $A^3$ are cheaper to calibrate. We observe that $A^3$-QK-SVD and $A^3$-OV-SVD and their simplified versions outperform CLOVER and Palu by a clear margin.

Table 2: A comparison of downstream task accuracy (↑).

| Model | CRatio | Method | ARC-c | BoolQ | Winogrande | GSM8k | MMLU | Avg. |
|---|---|---|---|---|---|---|---|---|
| LLaMA-2-7b (MHA-RoPE) | - | Original | 0.4829 | 0.7777 | 0.7498 | 0.1387 | 0.4582 | 0.5158 |
| | 10% | SVD-LLM | 0.3882 | 0.6749 | 0.6803 | 0.0129 | 0.3477 | 0.4166 |
| | | A³ | **0.4761** | **0.7330** | **0.7435** | **0.1130** | **0.4398** | **0.4960** |
| | 20% | SVD-LLM | 0.3139 | 0.6602 | 0.6464 | 0.0045 | 0.3119 | 0.3837 |
| | | A³ | **0.4369** | **0.7174** | **0.7072** | **0.0751** | **0.3979** | **0.4621** |
| LLaMA-2-13b (MHA-RoPE) | - | Original | 0.5538 | 0.8086 | 0.7711 | 0.2343 | 0.5513 | 0.5774 |
| | 10% | SVD-LLM | 0.4206 | **0.8061** | 0.7308 | 0.0902 | 0.4772 | 0.5000 |
| | | A³ | **0.5213** | 0.7865 | **0.7743** | **0.1971** | **0.5324** | **0.5560** |
| | 20% | SVD-LLM | 0.3472 | **0.7877** | 0.6898 | 0.0379 | 0.4318 | 0.4546 |
| | | A³ | **0.4727** | 0.7654 | **0.7364** | **0.1645** | **0.4804** | **0.5180** |
| LLaMA-3.1-8B (GQA-RoPE) | - | Original | 0.5401 | 0.8190 | 0.7822 | 0.4920 | 0.6535 | 0.6484 |
| | 10% | SVD-LLM | 0.3575 | 0.7458 | **0.7111** | 0.0447 | 0.4708 | 0.4603 |
| | | A³ | **0.4565** | **0.7884** | 0.7072 | **0.2388** | **0.5922** | **0.5500** |
| | 20% | SVD-LLM | 0.2534 | **0.6948** | **0.6440** | 0.0113 | 0.3604 | 0.3880 |
| | | A³ | **0.3345** | 0.6823 | 0.6417 | **0.0705** | **0.4649** | **0.4336** |
| LLaMA-3.1-70B (GQA-RoPE) | - | Original | 0.6536 | 0.8538 | 0.8445 | 0.8036 | 0.7864 | 0.7768 |
| | 10% | SVD-LLM | 0.5742 | 0.8401 | 0.8051 | 0.5087 | 0.7181 | 0.6797 |
| | | A³ | **0.6323** | **0.8532** | **0.8335** | **0.7453** | **0.7470** | **0.7508** |
| | 20% | SVD-LLM | **0.4957** | **0.8226** | **0.7727** | 0.3040 | **0.6620** | 0.6025 |
| | | A³ | 0.4667 | 0.8144 | 0.6875 | **0.4951** | 0.6145 | **0.6071** |

**Stronger KV Cache Compression**  @Reviewer n81d, @Reviewer gBeN Table 3 extends our attention ablation by comparing A³ with Clover and Palu across the full range of compression ratios on SlimPajama, C4, and WikiText-2, using both MPT-7B and MPT-30B. Although all three methods reduce parameters and KV-cache size by the **same** amount at a given compression ratio, their perplexity trends diverge significantly. Clover's performance degrades sharply, even at only 20% compression, its perplexity rises above 40 on the MPT-7B model. Palu is closer to A³, but it still underperforms by a large margin because its objective only reduces loss in the K and V projection outputs.

@Reviewer voSr In contrast, A³ consistently achieves the lowest perplexity across all datasets and model sizes. This performance gap widens at higher compression levels and with larger model sizes: on MPT-30B, A³ is the **only** method that stays below 50 perplexity at 80% compression, and on MPT-7B it is the only one that remains below 1k perplexity.

Table 3: A comparison of perplexity (↓) on WikiText2, C4, and SlimPajama. CRatio indicates compression ratio on both KV-Cache and Parameter count.

| MPT-7B | SlimPajama | | | C4 | | | Wikitext-2 | | |
|---|---|---|---|---|---|---|---|---|---|
| CRatio | Clover | Palu | A³ | Clover | Palu | A³ | Clover | Palu | A³ |
| 20% | 48.11 | 9.67 | **8.88** | 53.29 | 11.74 | **10.77** | 77.78 | 8.73 | **8.05** |
| 40% | 383 | 11.51 | **9.90** | 408 | 14.18 | **12.20** | 795 | 10.60 | **9.19** |
| 60% | 5397 | 25.73 | **15.34** | 4919 | 32.26 | **18.71** | 7895 | 25.09 | **15.58** |
| 80% | 15467 | 5270 | **388** | 11661 | 3210 | **373** | 14434 | 13714 | **849** |

| MPT-30B | SlimPajama | | | C4 | | | Wikitext-2 | | |
|---|---|---|---|---|---|---|---|---|---|
| CRatio | Clover | Palu | A³ | Clover | Palu | A³ | Clover | Palu | A³ |
| 20% | 11.52 | 7.91 | **7.71** | 14.53 | 9.87 | **9.59** | 13.07 | 7.04 | **6.73** |
| 40% | 18.00 | 8.99 | **8.33** | 22.43 | 11.30 | **10.44** | 23.47 | 8.40 | **7.40** |
| 60% | 54.97 | 15.59 | **11.52** | 70.65 | 18.91 | **14.22** | 95.45 | 18.88 | **11.28** |
| 80% | 779 | 211 | **37.09** | 732 | 253 | **42.85** | 1524 | 339 | **46.72** |

**Attention with RoPE** In Section 3.4 we propose using CUR approximation to solve Problem 1 for attention with RoPE, which follows a similar approach as $\mathtt{A^3\text{-}MLP}$ in Section 3.3. Here we compare against structured pruning baselines that can be adapted for this problem, including $\mathrm{abs}(w)$ and Wanda (Sun et al., 2023). $\mathrm{abs}(w)$ represents the classic pruning method that drops weights with smaller magnitudes, while Wanda sorts by the product between the weight magnitude and the average $l_2$-norm of the corresponding activation row [1]. Figure 4b illustrates $\Delta$PPL of LLaMA-2-7B on WikiText2, indicating the advantage of $\mathtt{A^3}$ over $\mathrm{abs}(w)$ and Wanda.

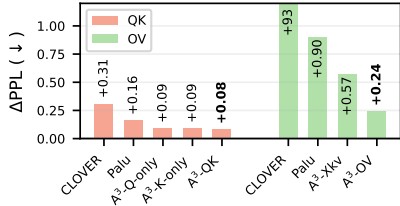

(a) QK and OV (MHA-NoPE).

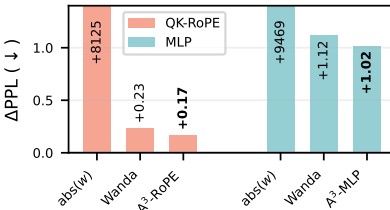

(b) QK-RoPE and MLP (MHA-RoPE).

Figure 4: Ablation study of $\mathtt{A^3}$ components. (a) $\mathtt{QK}$ and $\mathtt{OV}$ on MPT-7B. (b) $\mathtt{QK\text{-}RoPE}$ and $\mathtt{MLP}$ on LLaMA-2-7B.

## 5 DISCUSSION

In Section D, we showcase various applications of $\mathtt{A^3}$, highlighting its compatibility with quantization, lora fine-tuning and extensibility to mixed-rank allocation for additional performance gains. We find $\mathtt{A^3}$ is orthogonal to weight-only quantization methods like HQQ (Badri & Shaji, 2023) and a simple mixed-rank $\mathtt{A^3}$ outperforms ASVD and $\mathtt{SVD\text{-}LLM}$ v2. @Reviewer gBeN When combined with quantization, it yields a better Pareto frontier than using quantization alone at sub-4-bit setting. We also discuss the limitations of $\mathtt{A^3}$ in Section D, mainly caused by the sub-optimality of CUR decomposition, a compromise to RoPE. @Reviewer n81d Additionally, we provide empirical diagnostics that link the reductions in individual local objectives for $\mathtt{QK}$ and $\mathtt{OV}$ to the overall end-to-end perplexity. Finally, we explore the impact of calibration set selection on overall method performance, showing that a mixture of calibration datasets boosts accuracies on downstream tasks like Winogrande.

## 6 CONCLUSION

We propose $\mathtt{A^3}$, an analytical framework that decomposes the transformer into its core components, $\mathtt{QK}$, $\mathtt{OV}$, $\mathtt{MLP}$, and compresses them by minimizing their respective errors. This method reduces model size, KV cache, and FLOPs without runtime overhead, while achieving state-of-the-art performance.

---

[1]In the case of Equation (25), $\mathrm{abs}(w)$ drops columns (rows) by the column (row) sum of magnitudes of $\boldsymbol{W}_{q,i}$ ($\boldsymbol{W}_{k,i}^T$), while Wanda assumes non-diagonal elements in $\boldsymbol{R}_{\mathbb{X}_q \mathbb{X}_q}$ and $\boldsymbol{R}_{\mathbb{X}_{kv} \mathbb{X}_{kv}}$ are all zeros.

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

# A NOTATIONS

Table 4 includes the notations of matrices and vectors and Table 5 summarizes the notations of dimensions in this paper.

Table 4: Notation of matrices and vectors in this paper.

| Notation | Description |
|---|---|
| $\boldsymbol{R}_{\mathbb{XX}}$ | The autocorrelation matrices of $\boldsymbol{X} \in \mathbb{R}^{l \times d}$ computed as $\frac{1}{l}\boldsymbol{X}^T\boldsymbol{X}$ |
| $\boldsymbol{R}_{\mathbb{XX}}^{\frac{1}{2}}$ | The corresponding unique symmetric matrix square roots of $\boldsymbol{R}_{\mathbb{XX}}$ |
| $\boldsymbol{q}_{\mathrm{q},i}$ | An input row vector to qery projection of $i$-th head |
| $\boldsymbol{k}_{\mathrm{k},i}$ | An input row vector to key/value projection of $i$-th head |
| $\boldsymbol{X}_q$ | Input activation to the query layer |
| $\boldsymbol{X}_{\mathrm{kv}}$ | Input activation to the key/value layer |
| $\boldsymbol{W}_{q,i}$ | Weight of qery projection of $i$-th head |
| $\boldsymbol{W}_{k,i}$ | Weight of key projection of $i$-th head |
| $\boldsymbol{W}_{\mathrm{qk},i}$ | Fused weight of query/key projection of $i$-th head |
| $\widetilde{\boldsymbol{W}}_{\mathrm{qk},i}$ | Low-rank approximation of $\boldsymbol{W}_{\mathrm{qk},i}$ |
| $\widetilde{\boldsymbol{W}}_{\mathrm{q},i}$ | Approximated $\boldsymbol{W}_{q,i}$, left low-rank matrix of $\widetilde{\boldsymbol{W}}_{\mathrm{qk},i}$ |
| $\widetilde{\boldsymbol{W}}_{\mathrm{k},i}$ | Approximated $\boldsymbol{W}_{k,i}$, right low-rank matrix of $\widetilde{\boldsymbol{W}}_{\mathrm{qk},i}$ |
| $\boldsymbol{Q}_i$ | Query of $i$-th head |
| $\boldsymbol{K}_i$ | Key of $i$-th head |
| $a_i$ | A single attention score of $i$-th head |
| $\boldsymbol{A}_i$ | Pre-softmax attention score of $i$-th head |
| $\boldsymbol{A}'_i$ | Post-softmax attention score of $i$-th head |
| $\boldsymbol{W}_{v,i}$ | Weight of value projection of $i$-th head |
| $\boldsymbol{W}_{o,i}$ | Weight of output projection of $i$-th head |
| $\boldsymbol{W}_{\mathrm{vo},i}$ | Fused weight of value/output projection of $i$-th head |
| $\widetilde{\boldsymbol{W}}_{\mathrm{vo},i}$ | Low-rank approximation of $\boldsymbol{W}_{\mathrm{vo},i}$ |
| $\boldsymbol{o}$ | A row of attention output |
| $\boldsymbol{o}_i$ | A row of attention output $i$-th head |
| $\widetilde{\boldsymbol{o}}$ | Low-rank approximation of $\boldsymbol{o}$ |
| $\boldsymbol{O}$ | Attention output matrix |
| $\boldsymbol{x}_{\mathrm{d}}$ | An input row vector of down projection layer in MLP |
| $\widetilde{\boldsymbol{x}}_{\mathrm{d}}$ | An input row vector of down projection layer in MLP after low-rank approximation |
| $\boldsymbol{y}_{\mathrm{mlp}}$ | Output vectors of down projection layer in MLP after low-rank approximation |
| $\boldsymbol{X}_{\mathrm{mlp}}$ | Input of MLP in transformer |
| $\boldsymbol{X}_{\mathrm{d}}$ | Input of down projection layer in MLP |
| $\boldsymbol{Y}_{\mathrm{d}}$ | Output of gate projection layer in MLP |
| $\boldsymbol{Y}_{\mathrm{u}}$ | Output of up projection layer in MLP |
| $\boldsymbol{Y}_{\mathrm{mlp}}$ | Output of down projection layer in MLP |
| $\boldsymbol{W}_{\mathrm{u}}$ | Weight of up projection layer in MLP |
| $\boldsymbol{W}_{\mathrm{d}}$ | Weight of down projection layer in MLP |
| $\boldsymbol{W}_{\mathrm{g}}$ | Weight of gate projection layer in MLP |
| $\widetilde{\boldsymbol{W}}_{\mathrm{d}}$ | Weight of up projection layer in low-rank approximated MLP |
| $\widetilde{\boldsymbol{W}}_{\mathrm{d}}$ | Weight of down projection layer in low-rank approximated MLP |
| $\widetilde{\boldsymbol{W}}_{\mathrm{d}}$ | Weight of gate projection layer in low-rank approximated MLP |
| $\boldsymbol{r}_i$ | The $i$-th column of $\boldsymbol{R}_{\mathbb{X}_{\mathrm{d}}\mathbb{X}_{\mathrm{d}}}^{\frac{1}{2}}$ |
| $\boldsymbol{w}_i$ | The $i$-th row of $\boldsymbol{W}_{\mathrm{d}}$ |

Table 5: Notation of dimensions in this paper.

| Notation | Description |
|---|---|
| $l_\mathrm{q}$ | Query sequence length |
| $l_\mathrm{kv}$ | Key and value sequence length |
| $l_\mathrm{down}$ | Down layer sequence length |
| $d_\mathrm{m}$ | Model hidden size |
| $h_\mathrm{q}$ | Number of attention (query) heads |
| $h_\mathrm{kv}$ | Number of key and value heads |
| $g := \lfloor h_\mathrm{q}/h_\mathrm{kv} \rfloor$ | Number of query heads per key/value head in GQA |
| $d_\mathrm{vo}$ | Head dimension shared by value and head output projection |
| $d_\mathrm{qk}$ | Head dimension shared by query and key |
| $d_\mathrm{inter}$ | Intermediate size of FFN |

# B  DERIVATIONS FOR $\mathtt{A}^3$

## B.1  $\mathtt{A}^3$-QK

### B.1.1  EQUIVALENT OBJECTIVE

Here we provide the full derivation for Lemma 1 from Problem 1:

$$
\begin{aligned}
&\mathrm{argmin}_{\widetilde{\boldsymbol{W}}_{\mathrm{qk},i}} \|\boldsymbol{X}_\mathrm{q}(\boldsymbol{W}_{\mathrm{qk},i} - \widetilde{\boldsymbol{W}}_{\mathrm{qk},i})\boldsymbol{X}_\mathrm{kv}{}^T\|_F^2 \quad \text{s.t.} \quad \mathrm{rank}(\widetilde{\boldsymbol{W}}_{\mathrm{qk},i}) = r \\
&\Rightarrow \mathrm{argmin}_{\widetilde{\boldsymbol{W}}_{\mathrm{qk},i}} \|\boldsymbol{R}_{\mathbb{X}_\mathrm{q}\mathbb{X}_\mathrm{q}}^{\frac{1}{2}}(\boldsymbol{W}_{\mathrm{qk},i} - \widetilde{\boldsymbol{W}}_{\mathrm{qk},i})\boldsymbol{R}_{\mathbb{X}_\mathrm{kv}\mathbb{X}_\mathrm{kv}}^{\frac{1}{2}}\|_F^2 .
\end{aligned}
\tag{26}
$$

We begin with the right-hand side (RHS) of Equation (26). For clarity, we define some intermediate variables:

$$
\begin{aligned}
&\|\boldsymbol{X}_\mathrm{q}(\boldsymbol{W}_{\mathrm{qk},i} - \widetilde{\boldsymbol{W}}_{\mathrm{qk},i})\boldsymbol{X}_\mathrm{kv}{}^T\|_F^2 \\
&= Tr(\boldsymbol{X}_{kv}(\boldsymbol{W}_{\mathrm{qk},i} - \widetilde{\boldsymbol{W}}_{\mathrm{qk},i})^T \boldsymbol{X}_q^T \boldsymbol{X}_q (\boldsymbol{W}_{\mathrm{qk},i} - \widetilde{\boldsymbol{W}}_{\mathrm{qk},i})\boldsymbol{X}_{kv}^T) \\
&= Tr(\boldsymbol{X}_{kv}^T \boldsymbol{X}_{kv}(\boldsymbol{W}_{\mathrm{qk},i} - \widetilde{\boldsymbol{W}}_{\mathrm{qk},i})^T \boldsymbol{X}_q^T \boldsymbol{X}_q (\boldsymbol{W}_{\mathrm{qk},i} - \widetilde{\boldsymbol{W}}_{\mathrm{qk},i})) ,
\end{aligned}
\tag{27}
$$

If we assign $\boldsymbol{R}_{\mathbb{X}_\mathrm{kv}\mathbb{X}_\mathrm{kv}} = \frac{1}{l_\mathrm{kv}}\boldsymbol{X}_{kv}^T \boldsymbol{X}_{kv}$, and $\boldsymbol{R}_{\mathbb{X}_\mathrm{q}\mathbb{X}_\mathrm{q}} = \frac{1}{l_\mathrm{q}}\boldsymbol{X}_q^T \boldsymbol{X}_q$,

$$
\begin{aligned}
LHS &= Tr(l_\mathrm{kv} l_\mathrm{q} \boldsymbol{R}_{\mathbb{X}_\mathrm{kv}\mathbb{X}_\mathrm{kv}}((\boldsymbol{W}_{\mathrm{qk},i} - \widetilde{\boldsymbol{W}}_{\mathrm{qk},i})^T \boldsymbol{R}_{\mathbb{X}_\mathrm{q}\mathbb{X}_\mathrm{q}}((\boldsymbol{W}_{\mathrm{qk},i} - \widetilde{\boldsymbol{W}}_{\mathrm{qk},i})) \\
&= l_\mathrm{kv} l_\mathrm{q} \|\boldsymbol{R}_{\mathbb{X}_\mathrm{kv}\mathbb{X}_\mathrm{kv}}^{\frac{1}{2}}(\boldsymbol{W}_{\mathrm{qk},i} - \widetilde{\boldsymbol{W}}_{\mathrm{qk},i})\boldsymbol{R}_{\mathbb{X}_\mathrm{q}\mathbb{X}_\mathrm{q}}^{\frac{1}{2}}\|_F^2 \\
&= \|\boldsymbol{R}_{\mathbb{X}_\mathrm{kv}\mathbb{X}_\mathrm{kv}}^{\frac{1}{2}}(\boldsymbol{W}_{\mathrm{qk},i} - \widetilde{\boldsymbol{W}}_{\mathrm{qk},i})\boldsymbol{R}_{\mathbb{X}_\mathrm{q}\mathbb{X}_\mathrm{q}}^{\frac{1}{2}}\|_F^2 .
\end{aligned}
\tag{28}
$$

the positive $l_\mathrm{q}l_\mathrm{kv}$ can be dropped since they do not affect the minimizer.

### B.1.2  ANALYTICAL SOLUTION

Here we provide the proof of Theorem 2:

*Proof.* We continue with Lemma 1.

$$
\begin{aligned}
&\mathrm{argmin}_{\widetilde{\boldsymbol{W}}_{\mathrm{qk},i}} \|\boldsymbol{R}_{\mathbb{X}_\mathrm{q}\mathbb{X}_\mathrm{q}}^{\frac{1}{2}}(\boldsymbol{W}_{\mathrm{qk},i} - \widetilde{\boldsymbol{W}}_{\mathrm{qk},i})\boldsymbol{R}_{\mathbb{X}_\mathrm{kv}\mathbb{X}_\mathrm{kv}}^{\frac{1}{2}}\|_F^2 \quad \text{s.t.} \quad \mathrm{rank}(\widetilde{\boldsymbol{W}}_{\mathrm{qk},i}) = r \\
&\Rightarrow \mathrm{argmin}_{\widetilde{\boldsymbol{W}}_{\mathrm{qk},i}} \|\boldsymbol{R}_{\mathbb{X}_\mathrm{q}\mathbb{X}_\mathrm{q}}^{\frac{1}{2}}\boldsymbol{W}_{\mathrm{qk},i}\boldsymbol{R}_{\mathbb{X}_\mathrm{kv}\mathbb{X}_\mathrm{kv}}^{\frac{1}{2}} - \boldsymbol{R}_{\mathbb{X}_\mathrm{q}\mathbb{X}_\mathrm{q}}^{\frac{1}{2}}\widetilde{\boldsymbol{W}}_{\mathrm{qk},i}\boldsymbol{R}_{\mathbb{X}_\mathrm{kv}\mathbb{X}_\mathrm{kv}}^{\frac{1}{2}}\|_F^2 .
\end{aligned}
\tag{29}
$$

Note that multiplication by the invertible matrix $\boldsymbol{R}_{\mathbb{X}_\mathrm{q}\mathbb{X}_\mathrm{q}}^{\frac{1}{2}}$ and $\boldsymbol{R}_{\mathbb{X}_\mathrm{kv}\mathbb{X}_\mathrm{kv}}^{\frac{1}{2}}$ does not change the rank of the matrix $\boldsymbol{W}_{\mathrm{qk},i}$. According to the Eckart-Young-Mirsky theorem (Eckart & Young, 1936b), the optimal rank r approximation to $(\boldsymbol{R}_{\mathbb{X}_\mathrm{q}\mathbb{X}_\mathrm{q}}^{\frac{1}{2}}\boldsymbol{W}_{\mathrm{qk},i}\boldsymbol{R}_{\mathbb{X}_\mathrm{kv}\mathbb{X}_\mathrm{kv}}^{\frac{1}{2}})$ is the truncated SVD of $(\boldsymbol{R}_{\mathbb{X}_\mathrm{q}\mathbb{X}_\mathrm{q}}^{\frac{1}{2}}\boldsymbol{W}_{\mathrm{qk},i}\boldsymbol{R}_{\mathbb{X}_\mathrm{kv}\mathbb{X}_\mathrm{kv}}^{\frac{1}{2}})$:

$$(\boldsymbol{R}_{\mathbb{X}_q\mathbb{X}_q}^{\frac{1}{2}}\boldsymbol{W}_{\text{qk},i}\boldsymbol{R}_{\mathbb{X}_{\text{kv}}\mathbb{X}_{\text{kv}}}^{\frac{1}{2}})_r = \boldsymbol{U}_{:,:r}\boldsymbol{\Sigma}_{:r,:r}\boldsymbol{V}_{:r,:}^T \,, \tag{30}$$

where $\boldsymbol{U}\boldsymbol{\Sigma}\boldsymbol{V}^T = \text{SVD}\left(\boldsymbol{R}_{\mathbb{X}_q\mathbb{X}_q}^{\frac{1}{2}}\boldsymbol{W}_{\text{qk},i}\boldsymbol{R}_{\mathbb{X}_{\text{kv}}\mathbb{X}_{\text{kv}}}^{\frac{1}{2}}\right)$. Thus the optimal rank-k solution to $\widetilde{\boldsymbol{W}}_{\text{qk},i}$ is:

$$\widetilde{\boldsymbol{W}}_{\text{qk},i} = \left(\boldsymbol{R}_{\mathbb{X}_q\mathbb{X}_q}^{\frac{1}{2}}\right)^{-1}\text{SVD}_r\left(\boldsymbol{R}_{\mathbb{X}_q\mathbb{X}_q}^{\frac{1}{2}}\boldsymbol{W}_{\text{qk},i}\boldsymbol{R}_{\mathbb{X}_{\text{kv}}\mathbb{X}_{\text{kv}}}^{\frac{1}{2}}\right)\left(\boldsymbol{R}_{\mathbb{X}_{\text{kv}}\mathbb{X}_{\text{kv}}}^{\frac{1}{2}}\right)^{-1} \,. \tag{31}$$

$\square$

## B.2   A$^3$-OV

### B.2.1   EQUIVALENT OBJECTIVE

Similarly, we provide the derivation of the equivalent objective in Problem 2:

$$\begin{aligned}
&\text{argmin}_{\widetilde{\boldsymbol{W}}_{\text{vo},i}}\|\boldsymbol{P}_i(\boldsymbol{W}_{\text{vo},i} - \widetilde{\boldsymbol{W}}_{\text{vo},i})\|_F^2 \quad \text{s.t.} \quad \text{rank}(\widetilde{\boldsymbol{W}}_{\text{vo},i}) = r \\
&\Rightarrow \text{argmin}_{\widetilde{\boldsymbol{W}}_{\text{vo},i}}\|\boldsymbol{R}_{\mathbb{X}_{\boldsymbol{p}_i}\mathbb{X}_{\boldsymbol{p}_i}}^{\frac{1}{2}}(\boldsymbol{W}_{\text{vo},i} - \widetilde{\boldsymbol{W}}_{\text{vo},i})\|_F^2 \,.
\end{aligned} \tag{32}$$

*Proof.* We begin with the right-hand side (RHS) of Equation (32).

$$\begin{aligned}
\|\boldsymbol{O}_i - \tilde{\boldsymbol{O}}_i\|_F^2 &= \|\boldsymbol{P}_i(\boldsymbol{W}_{\text{vo},i} - \widetilde{\boldsymbol{W}}_{\text{vo},i})\|_F^2 \\
&= Tr((\boldsymbol{W}_{\text{vo},i} - \widetilde{\boldsymbol{W}}_{\text{vo},i})^T\boldsymbol{P}_i^T\boldsymbol{P}_i(\boldsymbol{W}_{\text{vo},i} - \widetilde{\boldsymbol{W}}_{\text{vo},i})) \,,
\end{aligned} \tag{33}$$

If we assign $\boldsymbol{R}_{\mathbb{X}_{\boldsymbol{p}_i}\mathbb{X}_{\boldsymbol{p}_i}} = \frac{1}{l_q}\boldsymbol{P}_i^T\boldsymbol{P}_i$,

$$\begin{aligned}
LHS &= Tr(l_q(\boldsymbol{W}_{\text{vo},i} - \widetilde{\boldsymbol{W}}_{\text{vo},i})^T\boldsymbol{R}_{\mathbb{X}_{\boldsymbol{p}_i}\mathbb{X}_{\boldsymbol{p}_i}}(\boldsymbol{W}_{\text{vo},i} - \widetilde{\boldsymbol{W}}_{\text{vo},i})) \\
&= l_q\|\boldsymbol{R}_{\mathbb{X}_{\boldsymbol{p}_i}\mathbb{X}_{\boldsymbol{p}_i}}^{1/2}(\boldsymbol{W}_{\text{vo},i} - \widetilde{\boldsymbol{W}}_{\text{vo},i})\|_F^2 \\
&= \|\boldsymbol{R}_{\mathbb{X}_{\boldsymbol{p}_i}\mathbb{X}_{\boldsymbol{p}_i}}^{1/2}(\boldsymbol{W}_{\text{vo},i} - \widetilde{\boldsymbol{W}}_{\text{vo},i})\|_F^2 \,.
\end{aligned} \tag{34}$$

the positive $l_q$ can be dropped since they do not affect the minimizer.   $\square$

### B.2.2   ANALYTICAL SOLUTION

Here we provide the proof of Theorem 3.

*Proof.* We continue with Equation 34:

$$\begin{aligned}
&\text{argmin}_{\widetilde{\boldsymbol{W}}_{\text{vo},i}}\|\boldsymbol{P}_i(\boldsymbol{W}_{\text{vo},i} - \widetilde{\boldsymbol{W}}_{\text{vo},i})\|_F^2 \quad \text{s.t.} \quad \text{rank}(\widetilde{\boldsymbol{W}}_{\text{vo},i}) = r \\
&\Rightarrow \text{argmin}_{\widetilde{\boldsymbol{W}}_{\text{vo},i}}\|\boldsymbol{R}_{\mathbb{X}_{\boldsymbol{p}_i}\mathbb{X}_{\boldsymbol{p}_i}}^{\frac{1}{2}}(\boldsymbol{W}_{\text{vo},i} - \widetilde{\boldsymbol{W}}_{\text{vo},i})\|_F^2 \,.
\end{aligned} \tag{35}$$

Note that multiplication by the invertible matrix $\boldsymbol{R}_{\mathbb{X}_{\boldsymbol{p}_i}\mathbb{X}_{\boldsymbol{p}_i}}$ does not change the rank of the matrix $\boldsymbol{W}_{\text{vo},i}$. According to the Eckart-Young-Mirsky theorem (Eckart & Young, 1936b), the optimal rank $r$ approximation to $(\boldsymbol{R}_{\mathbb{X}_{\boldsymbol{p}_i}\mathbb{X}_{\boldsymbol{p}_i}}^{\frac{1}{2}}\boldsymbol{W}_{\text{vo},i})$ is the truncated SVD of $(\boldsymbol{R}_{\mathbb{X}_{\boldsymbol{p}_i}\mathbb{X}_{\boldsymbol{p}_i}}^{\frac{1}{2}}\boldsymbol{W}_{\text{vo},i})$:

$$(\boldsymbol{R}_{\mathbb{X}_{\boldsymbol{p}_i}\mathbb{X}_{\boldsymbol{p}_i}}^{\frac{1}{2}}\boldsymbol{W}_{\text{vo},i})_r = \boldsymbol{U}_{:,:r}\boldsymbol{\Sigma}_{:r,:r}\boldsymbol{V}_{:r,:}^T \,, \tag{36}$$

where $\boldsymbol{U}\boldsymbol{\Sigma}\boldsymbol{V}^T = \text{SVD}\left(\boldsymbol{R}_{\mathbb{X}_{\boldsymbol{p}_i}\mathbb{X}_{\boldsymbol{p}_i}}^{\frac{1}{2}}\boldsymbol{W}_{\text{vo},i}\right)$. Thus the optimal rank-k solution to $\widetilde{\boldsymbol{W}}_{\text{vo},i}$ is:

$$\widetilde{\boldsymbol{W}}_{\text{vo},i} = \left(\boldsymbol{R}_{\mathbb{X}_q\mathbb{X}_q}^{\frac{1}{2}}\right)^{-1}\text{SVD}_r\left(\boldsymbol{R}_{\mathbb{X}_{\boldsymbol{p}_i}\mathbb{X}_{\boldsymbol{p}_i}}^{\frac{1}{2}}\boldsymbol{W}_{\text{vo},i}\right) \,. \tag{37}$$

$\square$

### B.2.3  ALTERNATIVE SOLUTION TO A³-OV

Here we elaborate on the alternative solution to Problem 2 by directly minimizing $\|\boldsymbol{O} - \widetilde{\boldsymbol{O}}\|_F^2$ through matrix stacking. With matrix stacking, we can write the overall attention output as two matrix multiplications:

$$\boldsymbol{O} = \sum_{i=1}^{h_q} \boldsymbol{O}_i = \sum_{i=1}^{h_q} \mathbf{P}_i \boldsymbol{W}_{\text{vo},i} = \begin{bmatrix} \boldsymbol{P}_1 & \boldsymbol{P}_2 & \dots \boldsymbol{P}_{h_q} \end{bmatrix} \begin{bmatrix} \boldsymbol{W}_{\text{vo},1} \\ \boldsymbol{W}_{\text{vo},2} \\ \vdots \\ \boldsymbol{W}_{\text{vo},h_q} \end{bmatrix} = \boldsymbol{P}_{\text{cat}} \boldsymbol{W}_{\text{vo,cat}} \ , \qquad (38)$$

where $\boldsymbol{P}_{\text{cat}} \in \mathbb{R}^{l_q \times d_m d_{h_{kv}}}, \boldsymbol{W}_{\text{vo,cat}} \in \mathbb{R}^{d_m d_{h_{kv}} \times d_m}$ denote the concatenated attention score weighted values and the fused value/output projection.

The overall term $\boldsymbol{O}$ takes the form of a linear layer $\boldsymbol{O} = \boldsymbol{P}_{\text{cat}} \boldsymbol{W}_{\text{vo,cat}}$. If $\widetilde{\boldsymbol{O}} = \boldsymbol{P}_{\text{cat}} \widetilde{\boldsymbol{W}}_{\text{vo,cat}}$ denotes the approximated $\boldsymbol{o}$, the optimal solution to $\widetilde{\boldsymbol{W}}_{\text{vo,cat}}$ is already given by Equation (8).

**Problem 4** (Minimization of overall attention output error). Given a pretrained Transformer layer, the attention output of OV component $\boldsymbol{O} = \boldsymbol{P}_{\text{cat}} \boldsymbol{W}_{\text{vo,cat}}$ and its approximated form $\widetilde{\boldsymbol{O}} = \boldsymbol{P}_{\text{cat}} \widetilde{\boldsymbol{W}}_{\text{vo,cat}}$, approximating the fused value/output projection by minimizing the output error is to minimize the following error:

$$\|\boldsymbol{P}_{\text{cat}}(\boldsymbol{W}_{\text{vo,cat}} - \widetilde{\boldsymbol{W}}_{\text{vo,cat}})\|_F^2 \quad \text{s.t.} \quad \text{rank}(\widetilde{\boldsymbol{W}}_{\text{vo,cat}}) = r \ . \qquad (39)$$

**Theorem 5** (A³-OV-overall for MHA-NoPE). *The optimal solution to Problem 4 is*

$$\widetilde{\boldsymbol{W}}_{vo,cat} = \left( \boldsymbol{R}_{\mathbb{X}_{\boldsymbol{p}_{cat}} \mathbb{X}_{\boldsymbol{p}_{cat}}}^{\frac{1}{2}} \right)^{-1} \text{SVD}_r ( \boldsymbol{R}_{\mathbb{X}_{\boldsymbol{p}_{cat}} \mathbb{X}_{\boldsymbol{p}_{cat}}}^{\frac{1}{2}} \boldsymbol{W}_{vo,cat} ) \ , \qquad (40)$$

*where $\boldsymbol{R}_{\mathbb{X}_{\boldsymbol{p}_{cat}} \mathbb{X}_{\boldsymbol{p}_{cat}}}$ is the autocorrelation matrix respect to $\boldsymbol{P}_{cat}$ and $\boldsymbol{R}_{\mathbb{X}_{\boldsymbol{p}_{cat}} \mathbb{X}_{\boldsymbol{p}_{cat}}}^{\frac{1}{2}}$ denotes its unique symmetric matrix square root.*

Similar to QK component, in practice we assign

$$\boldsymbol{L}_{vo} := \left( \boldsymbol{R}_{\mathbb{X}_{\boldsymbol{p}_{cat}} \mathbb{X}_{\boldsymbol{p}_{cat}}}^{\frac{1}{2}} \right)^{-1} \boldsymbol{U}_{:,:k}, \quad \boldsymbol{R}_{vo} := \boldsymbol{\Sigma}_{:k,:k} \boldsymbol{V}_{:k,:}^T \left( \boldsymbol{R}_{\mathbb{X}_{\boldsymbol{p}_{cat}} \mathbb{X}_{\boldsymbol{p}_{cat}}}^{\frac{1}{2}} \right)^{-1} \ ,$$

to get the approximated fused value/output weights with two low-rank matrices. In terms of attention head, $\boldsymbol{L}_{vo}$ can be viewed as concatenating all value heads with head dimension of $r$ together, and $\boldsymbol{R}_{vo}$ can be viewed as a shared output head across the values:

$$\boldsymbol{L}_{vo} := \begin{bmatrix} \boldsymbol{L}_{v,1} \\ \boldsymbol{L}_{v,2} \\ \vdots \\ \boldsymbol{L}_{v,h_q} \end{bmatrix} \ , \quad \boldsymbol{R}_{vo} = \boldsymbol{R}_o \ ,$$

where $\boldsymbol{L}_{v,i} \in \mathbb{R}^{d_m \times r}$ and $\boldsymbol{R}_o \in \mathbb{R}^{r \times d_m}$. Attention output can then be computed as follows:

$$\boldsymbol{O} = \sum_{i=1}^{h_q} \boldsymbol{P}_i \boldsymbol{X}_{kv} \boldsymbol{L}_{v,i} \boldsymbol{R}_o \ . \qquad (41)$$

Although this solution can theoretically save more parameters than minimizing the per-head attention output loss under the same model performance, it requires significantly more KV-cache storage, even exceeding that of the uncompressed model. This is because the shared rank $r$ across all heads typically needs to be larger than $d_{\text{vo}}$ of one head to maintain competitive performance. As a result, the KV-cache size increases by a factor of $\frac{r}{d_{\text{vo}}}$ compared to uncompressed models.

### B.3  A³-MLP

#### B.3.1  EQUIVALENT OBJECTIVE

Here we provide the derivation of Lemma 4 in Problem 3:

$$\text{argmin}_{\boldsymbol{U}=\text{diag}(u_1,u_2,\dots,u_{d_{\text{inter}}})} \|\boldsymbol{X}_d \boldsymbol{U} \boldsymbol{W}_d - \boldsymbol{X}_d \boldsymbol{W}_d\|_F^2 \quad \text{s.t.} \quad \text{rank}(\boldsymbol{U}) = r \ ,$$

$$\Rightarrow \text{argmin}_{\boldsymbol{U}=\text{diag}(u_1,u_2,\dots,u_{d_{\text{inter}}})} \|\boldsymbol{R}_{\mathbb{X}_d \mathbb{X}_d}^{\frac{1}{2}} \boldsymbol{U} \boldsymbol{W}_d - \boldsymbol{R}_{\mathbb{X}_d \mathbb{X}_d}^{\frac{1}{2}} \boldsymbol{W}_d\|_F^2 \ . \tag{42}$$

*Proof.* We begin with the right-hand side (RHS) of Equation (42). For clarity, we define some intermediate variables:

$$\boldsymbol{V} := (\boldsymbol{U}\boldsymbol{W}_{down} - \boldsymbol{W}_{down}) = \left[\boldsymbol{v}_1^T, \boldsymbol{v}_2^T, \ldots, \boldsymbol{v}_{d_{\text{inter}}}^T\right]^T, \quad \boldsymbol{x}_{down} = \begin{bmatrix} x_1 & x_2 & \ldots & x_{d_{\text{inter}}} \end{bmatrix}. \quad (43)$$

We continue by substituting Equation (43) to RHS of Equation (42):

$$\begin{aligned}
&\|\boldsymbol{X}_{\text{d}}\boldsymbol{U}\boldsymbol{W}_{\text{d}} - \boldsymbol{X}_{\text{d}}\boldsymbol{W}_{\text{d}}\|_F^2 \\
&= \|\boldsymbol{X}_{\text{d}}\boldsymbol{V}\|_F^2 \\
&= \text{Tr}((\boldsymbol{X}_{\text{d}}\boldsymbol{V})^T \boldsymbol{X}_{\text{d}}\boldsymbol{V}) \\
&= \text{Tr}(\boldsymbol{V}^T \boldsymbol{X}_{\text{d}}^T \boldsymbol{X}_{\text{d}}\boldsymbol{V}) \\
&= \text{Tr}(\boldsymbol{X}_{\text{d}}^T \boldsymbol{X}_{\text{d}}\boldsymbol{V}\boldsymbol{V}^T),
\end{aligned} \quad (44)$$

If we assign $\boldsymbol{R}_{\mathbb{X}_d \mathbb{X}_d} = \frac{1}{l_{\text{down}}}\boldsymbol{X}_{\text{d}}^T \boldsymbol{X}_{\text{d}}$,

$$\begin{aligned}
LHS &= \text{Tr}(\boldsymbol{R}_{\mathbb{X}_{\text{d}}\mathbb{X}_{\text{d}}}\boldsymbol{V}\boldsymbol{V}^T) \\
&= \text{Tr}(l_{\text{down}}\boldsymbol{R}_{\mathbb{X}_{\text{d}}\mathbb{X}_{\text{d}}}^{\frac{1}{2}}\boldsymbol{V}\boldsymbol{V}^T(\boldsymbol{R}_{\mathbb{X}_{\text{d}}\mathbb{X}_{\text{d}}}^{\frac{1}{2}})^T) \\
&= l_{\text{down}}\|\boldsymbol{R}_{\mathbb{X}_{\text{d}}\mathbb{X}_{\text{d}}}^{\frac{1}{2}}\boldsymbol{V}\|_F^2 \\
&= l_{\text{down}}\|\boldsymbol{R}_{\mathbb{X}_{\text{d}}\mathbb{X}_{\text{d}}}^{\frac{1}{2}}\boldsymbol{U}\boldsymbol{W}_{\text{d}} - \boldsymbol{R}_{\mathbb{X}_{\text{d}}\mathbb{X}_{\text{d}}}^{\frac{1}{2}}\boldsymbol{W}_{\text{d}}\|_F^2 \\
&= \|\boldsymbol{R}_{\mathbb{X}_{\text{d}}\mathbb{X}_{\text{d}}}^{\frac{1}{2}}\boldsymbol{U}\boldsymbol{W}_{\text{d}} - \boldsymbol{R}_{\mathbb{X}_{\text{d}}\mathbb{X}_{\text{d}}}^{\frac{1}{2}}\boldsymbol{W}_{\text{d}}\|_F^2.
\end{aligned} \quad (45)$$

the positive $l_{\text{down}}$ can be dropped since they do not affect the minimizer. □

## B.4 EXTENDING $A^3$ FOR GQA

Similar to GQA's QK Component, we can apply joint SVD to the OV component. The concatenation is done along the second dimension:

$$\text{SVD}\left(\begin{bmatrix} \boldsymbol{R}_{\mathbb{X}_{\text{kv}}\mathbb{X}_{\text{kv}}}^{\frac{1}{2}}\widetilde{\boldsymbol{W}}_{\text{vo},1} & \ldots & \boldsymbol{R}_{\mathbb{X}_{\text{kv}}\mathbb{X}_{\text{kv}}}^{\frac{1}{2}}\widetilde{\boldsymbol{W}}_{\text{vo},g} \end{bmatrix}\right) = U\Sigma V^T. \quad (46)$$

Then we make the assignment below to build the shared value head weights $\widetilde{\boldsymbol{W}}_{\text{v, shared}}$ and the $i$-th approximated output head weights $\widetilde{\boldsymbol{W}}_{\text{o},i}$ for this group.

$$\widetilde{\boldsymbol{W}}_{\text{o},i} := V_{:r,\, id_{\text{m}}:(i+1)d_{\text{m}}}^T \left(\boldsymbol{R}_{\mathbb{X}_{\text{kv}}\mathbb{X}_{\text{kv}}}^{\frac{1}{2}}\right)^{-1}, \quad \widetilde{\boldsymbol{W}}_{\text{v, shared}} := U_{:,\,:r}\Sigma_{:r,\,:r}. \quad (47)$$

Note that in Equation (46) we use $\boldsymbol{R}_{\mathbb{X}_{\text{kv}}\mathbb{X}_{\text{kv}}}$ instead of $\boldsymbol{R}_{\mathbb{X}_{\boldsymbol{p}_i}\mathbb{X}_{\boldsymbol{p}_i}}$, because $\boldsymbol{R}_{\mathbb{X}_{\boldsymbol{p}_i}\mathbb{X}_{\boldsymbol{p}_i}}$ is head-specific, which prevents us from building a shared $\widetilde{\boldsymbol{W}}_{\text{v,shared}}$ for all heads in the same GQA group. We name these two methods $A^3$-QK-CR and $A^3$-OV-CR for GQA's QK and OV components respectively.

## B.5 EXTENDING $A^3$ FOR RoPE

Instead of sorting by the product of $l_2$-norm, $\lambda_i = \|\boldsymbol{L}_{:,\,i}\|_2^2 \cdot \|\boldsymbol{R}_{i,\,:}\|_2^2$, for $i = 0, 1, \ldots, d_{\text{qk}} - 1$, we sort by the sum of $\lambda_i$ of adjacent pairs:

$$\lambda_{2i} = \|\boldsymbol{L}_{:,\,2i}\|_2^2 \cdot \|\boldsymbol{R}_{2i,\,:}\|_2^2 + \|\boldsymbol{L}_{:,\,2i+1}\|_2^2 \cdot \|\boldsymbol{R}_{2i+1,\,:}\|_2^2 \quad \text{for } i = 0, 1, \ldots, \frac{d_{\text{qk}}}{2} - 1. \quad (48)$$

This will drop pairs of less important columns in $\boldsymbol{L}$ and rows in $\boldsymbol{R}$, as well as the corresponding pairs of RoPE frequencies. This requires a frequency index array for each QK pair, and indexing the RoPE constants at runtime. Usually the head dimension $d_{\text{qk}}$ is 64 or 128, so the RoPE frequency indices can be saved in a compact INT8 array. However, to achieve high throughput/low latency, a custom kernel is needed to fuse indexing and rotation together, which is out of the scope of this paper.

This RoPE extension can be combined with the GQA extension, which means the sorting in Equation (48) is done on the concatenated $\boldsymbol{L}$ and $\boldsymbol{R}$ matrices in Equation (22).

## C  DETAILED EXPERIMENT SETUP

**Calibration**  We concatenate the texts in SlimPajama and randomly sample 128 sequences of 2048 tokens for calibration. We only calibrate on WikiText-2 for Table 8. SlimPajama is a pretraining dataset of high-quality corpus, better capturing the statistics of auto-correlation than WikiText2. We calibrate the auto-correlation matrix using BF16 models, but accumulate the outer product in FP64.

**Approximation**  Since the autocorrelation matrix is symmetric and positive semi-definite, we used SVD to calculate its inverse and matrix square root, which improves the numerical stability. For GQA models, we also use `torch.svd_lowrank` instead of `torch.linalg.svd` for faster solving. @Reviewer voSr, @Reviewer n81d In cases where the autocorrelation matrices are ill-conditioned, we follow the approach of GPTQ (Frantar et al., 2022) and add a small damping term to the zero eigenvalues. In all of our experiments across different calibration datasets, the autocorrelation matrices were always invertible.

**Downstream evaluation**  We use 0-shot prompting for BoolQ and OpenBookQA, 5-shot for Winogrande, GSM8K, and MMLU, 25-shot for ARC-c. Other evaluation parameters are kept as the default provided by `lm-eval-harness`.

**@Reviewer gBeN Server specs**  We run all the experiments on two GPU boxes, one with two NVIDIA H100s, and the other with 8 NVIDIA A100s. In total, we spent around 1200 GPU hours on running $A^3$, and 800 hours on baselines. Specifically, for MHA models, most of the GPU hours were spent on calibration and VO solving. For GQA models, `torch.svd_lowrank` speeds up the VO solving, with most of the GPU hours on calibration and FFN solving. For the ASVD baseline, most of the GPU hours were on the mixed-rank search, while other baselines took most of the time on calibration and approximation. Since all the calibration, approximation, and evaluation were run on GPUs, our experiments were not bottlenecked by CPUs.

## D  DISCUSSION

**Quantization compatibility**  Here we show that $A^3$ can be combined together with quantization. Figure 5 presents the perplexity of quantized LLaMA-3.1-8B with HQQ 4-bit quantization, before and after applying $A^3$. The small amount of extra model performance degradation caused by $A^3$ indicates its orthogonality to quantization.

@Reviewer gBeN For extreme compression regimes, combining low-rank methods with quantization creates a continuous spectrum of compression levels between discrete quantization points and can yield a way better Pareto frontier. As shown in Figure 6, sub-3-bit quantization alone destabilizes the model, whereas $A^3$ + quantization provides a significantly better accuracy vs compression trade-off than quantization along.

**Mixed-rank allocation**  SVD-LLM v2 (Wang et al., 2025) and ASVD(Yuan et al., 2023) have demonstrated that different layers exhibit varying sensitivity to rank reduction. Here we show that $A^3$ can also benefit from mixed-rank allocation, achieving performance gain with minimal effort. Specifically, we conduct a simple search over rank allocations for each decoder layer in LLaMA-7b. As shown in Table 6, $A^3$-mixed outperforms both the uniform $A^3$ and other mixed-rank approaches.

**Limitation of CUR decomposition**  One limitation of $A^3$ is its reliance on CUR decomposition for RoPE-based

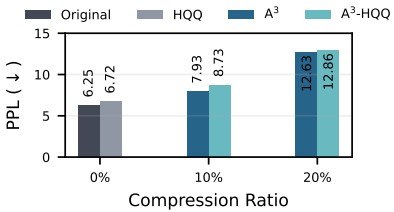

Figure 5: A comparison of perplexity ($\downarrow$) on WikiText-2 using quantization (HQQ, 4 bits) for both the original and $A^3$-applied LLaMA-3.1-8B.

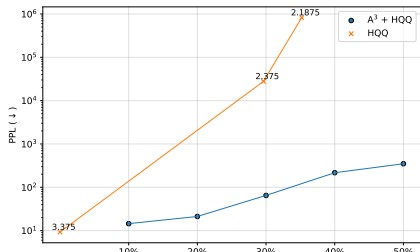

Figure 6: A comparison of perplexity ($\downarrow$) on WikiText-2 for HQQ + $A^3$ relative to applying HQQ alone in the sub–4-bit setting on LLaMA-3.1-8B.

attention and MLP, which does not guarantee an optimal solution like SVD. When targeting a small compression ratio (*e.g.*, CRatio=10%), CUR decomposition provides a good trade-off between performance and compression. As the compression ratio increases, its performance degrades much faster than SVD and is eventually surpassed by SVD-based approaches. However, we argue that even for SVD-based approaches, the model performance under a compression ratio larger than 10% is already very poor. For example, the C4 perplexity of LLaMA-3.1-70B with CRatio=20% is 13.77 for `SVD-LLM`, which is even higher than the original LLaMA-3.1-8B. A retraining is needed in this case, but is out of the scope of this paper.

**Choice of calibration datasets** We compared the model performance when calibrating on SlimPajama and WikiText-2. We find calibrating on SlimPajama gives closer perplexities on Wikitext-2 and C4, regardless of the compression level (Appendix Table 7). However, calibrating on WikiText-2 has a widening perplexity gap between WikiText-2 and C4 as the compression ratio increased, especially for `SVD-LLM`, which potentially indicates overfitting. We hypothesize that this may contribute to cases where `SVD-LLM` appears to perform better on particular downstream tasks, like Winogrande, in Table 2. To validate this, we used a more diverse calibration set with samples from SlimPajama and PTB in Table 8. The results show that with this mixture of calibration sets, $A^3$ achieves a higher accuracy than `SVD-LLM` on Winogrande.

Table 6: A comparison of perplexity ($\downarrow$) on WikiText-2 and C4 of LLaMA-7B under 20% mixed-rank compression rate.

| Method | WikiText-2 | C4 |
|---|---|---|
| Original | 5.68 | 7.65 |
| $A^3$ | 7.21 | 10.01 |
| `SVD-LLM` v2 | 10.53 | 13.00 |
| ASVD | 10.45 | 13.1 |
| $A^3$-mix | **7.11** | **9.86** |

Table 7: Performance of LLaMA-7b compressed by SVD-LLM and $A^3$ under different compression ratio using calibration data sampled from Slimpajama (our setting) and WikiText-2 datasets. The performance are reported by the average and difference in perplexity ($\downarrow$) of Wikitext-2 and C4 datasets.

| Method | 10% | | 20% | | 40% | |
|---|---|---|---|---|---|---|
| | Avg | $|\Delta|$ | Avg | $|\Delta|$ | Avg | $|\Delta|$ |
| `SVD-LLM` (SlimPajama) | 9.50 | 2.54 | 9.50 | 2.54 | 30.84 | 1.00 |
| $A^3$ (SlimPajama) | 7.24 | 2.24 | 8.52 | 2.79 | 25.22 | 9.83 |
| `SVD-LLM` (WikiText-2) | 9.62 | 5.07 | 11.89 | 7.90 | 44.58 | 61.69 |
| $A^3$ (WikiText-2) | 7.24 | 2.25 | 8.50 | 3.68 | 12.82 | 18.95 |

Table 8: Performance evaluation of LLaMA-3.1-8b (20% compression) using `SVD-LLM` and $A^3$ with two calibration datasets: SlimPajama and a 50/50 SlimPajama-PTB mixture. Metrics include perplexity ($\downarrow$) of Wikitext-2, C4 and slimPajama, and accuracy ($\uparrow$) of BoolQ, Winogrande, ARC-c (with their average).

| Method | WikiText-2 | C4 | SlimPajama | BoolQ | Winogrande | ARC-c | Avg. |
|---|---|---|---|---|---|---|---|
| `SVD-LLM` (SlimPajama) | 42.28 | 33.6 | 27.44 | 0.6948 | 0.644 | 0.2534 | 0.5307 |
| $A^3$ (SlimPajama) | 11.36 | 17.87 | 13.58 | 0.6823 | 0.6417 | 0.3345 | 0.5528 |
| `SVD-LLM` (SlimPajama+PTB) | 36.76 | 36.62 | 31.79 | 0.7349 | 0.6717 | 0.2671 | 0.5579 |
| $A^3$ (SlimPajama+PTB) | 11.47 | 18.57 | 14.30 | 0.7220 | 0.6938 | 0.3524 | 0.5894 |

**Fine-tuning performance** Here we provide the $A^3$ performance on Wikitext-2 (PPL $\downarrow$) with simple LoRA fine-tuning setting. Following the `SVD-LLM` setup, we applied LoRA (rank 8) on $A^3$ with 50K Alpaca-cleaned samples over 2 epochs. Table 9 show $A^3$ benefits notably from even basic fine-tuning due to its strong initialization.

| Compression Ratio | $\mathtt{A}^3$ | $\mathtt{A}^3$ + Fine-Tuning |
|---|---|---|
| 20% | 7.22 (+1.73) | 6.94 (+1.45) |
| 40% | 32.04 (+24.31) | 10.53 (+5.04) |

Table 9: Comparison of $\mathtt{A}^3$ and $\mathtt{A}^3$ + Fine-Tuning across compression ratios for Llama-2-7b.

**Relation of local objective reduction to end-to-end perplexity**  @Reviewer n81d Here we provide a diagnostic analysis to better understand the gap between individual local objective reduction and their combined effect on end-to-end perplexity. Table 10 compares how locally compressing QK and OV impacts the global perplexity for two model architectures: MPT-7B and LLaMA-3.1-8B. For small compression ratios, the increase in perplexity is approximately equal to the sum of the individual contributions from QK and OV, particularly for standard $\mathtt{A}^3$-QK and $\mathtt{A}^3$-OV configurations without RoPE in MPT-7b. At larger compression ratios, the sum of contributions from QK and OV remains roughly on the same order of magnitude as the observed end-to-end perplexity.

| LLaMA-3.1-8B | 5% | 10% | 15% | 20% | 40% |
|---|---|---|---|---|---|
| QK | 0.07 | 0.16 | 0.32 | 0.56 | 13.28 |
| OV | 0.27 | 0.39 | 0.58 | 0.78 | 2.79 |
| QK + OV | 0.34 | 0.55 | 0.90 | 1.34 | 16.07 |
| Both | 0.35 | 0.59 | 1.00 | 1.58 | 25.07 |

| MPT-7B | 5% | 10% | 15% | 20% | 40% |
|---|---|---|---|---|---|
| QK | -0.004 | 0.005 | 0.040 | 0.092 | 0.75 |
| OV | 0.048 | 0.097 | 0.166 | 0.248 | 0.98 |
| QK + OV | 0.045 | 0.103 | 0.206 | 0.340 | 1.73 |
| Both | 0.044 | 0.102 | 0.197 | 0.313 | 1.52 |

Table 10: Perplexity changes under local and joint compression. Shows the effect of compressing QK and OV individually, their summed contribution (QK + OV), and **Both**, the end-to-end perplexity when both are compressed jointly, across different compression ratios for LLaMA-3.1-8B (top) and MPT-7B (bottom).

# E  RUNTIME ANALYSIS

## E.1  $\mathtt{A}^3$ COMPRESSION TRANSLATES TO GAINS IN RUNTIME PERFORMANCE

Here we provide more details for the runtime performance $\mathtt{A}^3$ including the rank, theoretical FLOPs for one decoder layer, throughput and peak allocated GPU memory across different compression ratio on 1 H100 GPU.

To compute theoretical FLOPs for a single decoder block during prefill, we sum attention, FFN, normalization, and residual costs:

$$\text{FLOPs}_{\text{total}} = \text{FLOPs}_{\text{attn}} + \text{FLOPs}_{\text{mlp}} + \text{FLOPs}_{\text{norm}} + \text{FLOPs}_{\text{resid}} \, ,$$

Attention ($Q, K, V, O$ projections + $QK^T$ + softmax + $AV$):

$$\text{FLOPs}_{\text{attn}} = 8BLHD + 4BL^2AD + BL^2A \, ,$$

Feedforward network (3 projections, each 2 FLOPs per multiply-add):

$$\text{FLOPs}_{\text{mlp}} = 6BLIH \, ,$$

Normalization and residuals (each counted twice):

$$\text{FLOPs}_{\text{norm}} = 2BLH \qquad \text{FLOPs}_{\text{resid}} = 2BLH \, ,$$

Final total:

$$\text{FLOPs}_{\text{total}} = 8BLHD + 4BL^2AD + BL^2A + 6BLIH + 4BLH \, .$$

Where: $B$ = batch size, $L$ = sequence length, $H$ = hidden size, $D$ = head dimension, $A$ = number of attention heads, $I$ = MLP intermediate size.

Since $\text{A}^3$ compresses the model by proportionally reducing both $D$ and $I$, the overall theoretical FLOPs reduction is approximately equal to the compression ratio, as many operations scale with $D$ and $I$. However, it is not exactly equal due to additional operations such as normalization, residual connections, and softmax. Table 11 highlights that $\text{A}^3$ compression ratio can directly translate to gains in runtime performance that closely align with the theoretical expectation without requiring specialized kernels.

| Compression | Ranks (qk/vo/mlp) | Method | Theoretical FLOPs | Throughput (token/s) | Peak Memory (MB) | Theoretical FLOPs | Speedup | Peak Memory |
|---|---|---|---|---|---|---|---|---|
| Original | 128/128/13824 | Eager | $2.77 \times 10^{12}$ | 7285 | 35004 | 1.00x | 1.00x | 1.00x |
| | | SDPA | | 12319 | 32917 | 1.00x | 1.69x | 0.94x |
| 20% | 128/128/13824 | Eager | $2.16 \times 10^{12}$ | 8077 | 28114 | 0.78x | 1.11x | 0.80x |
| | | SDPA | | 15096 | 26037 | 0.78x | 2.07x | 0.74x |
| 40% | 128/128/13824 | Eager | $1.56 \times 10^{12}$ | 9350 | 21336 | 0.56x | 1.28x | 0.61x |
| | | SDPA | | 20237 | 19270 | 0.56x | 2.78x | 0.55x |
| 60% | 128/128/13824 | Eager | $1.08 \times 10^{12}$ | 10405 | 16139 | 0.39x | 1.43x | 0.46x |
| | | SDPA | | 25554 | 14078 | 0.39x | 3.51x | 0.40x |

Table 11: Analysis of runtime performance in LLaMA-2-13b across varying compression ratios and methods.

### E.2 @REVIEWER N81D $\text{A}^3$ THROUGHPUT EVIDENCE AT SCALE

To evaluate the robustness of $\text{A}^3$'s throughput gains, Figure 7 shows throughput gains across a variety of settings, varying GPU type, batch size, model size, compression ratio, sequence length, and attention kernel. The experiments include:

- **GPU & Model Size:** Single A6000 (Llama-3.2-1B/Llama-3.2-3B/Llama-3.1-8B), Single H100 (Llama-3.2-3B/Llama-2-13B/Qwen3-32B)
- **Batch Size:** 1, 2, 4 for A6000; 1, 4, 8 for H100
- **Compression Ratio:** 20%, 40%
- **Sequence Length:** 1024, 2048
- **Attention Kernel:** eager, SDPA

Aligned with the analysis in Appendix E.1, $\text{A}^3$ consistently achieves speedups close to the theoretical gains, as it reduces the effective problem size without introducing overhead or additional kernel launches in SVD-LLM.

Larger models and SDPA attention benefit more from $\text{A}^3$, since the reduced head dimension not only compresses the linear layers but also decreases attention FLOPs.

## F OFFLINE COMPRESSION COST ANALYSIS

| Model | $\text{A}^3$-QK (min) | $\text{A}^3$-OV (min) | $\text{A}^3$-MLP (min) |
|---|---|---|---|
| LLaMA-7B MHA | 00:49 | 46:55 | 12:55 |
| LLaMA-8B GQA | 00:56 | 01:30 | 23:00 |

Table 12: Time (in minutes) for different model components.

Table 12 shows the compression time for both MHA (Llama-2-7b) and GQA (Llama-3-8b). Note that the compression is done offline before deployment and inference.

$\text{A}^3$-OV compression for $\text{A}^3$ on MHA models is more time-consuming because Equation 17 must be applied separately to each attention head. To ensure numerical stability, we use the weight truncation method from SVD-LLM-v2 (Wang et al., 2025), which involves two SVD operations per head: one to compute $R^{1/2}$, and another for the primary decomposition.

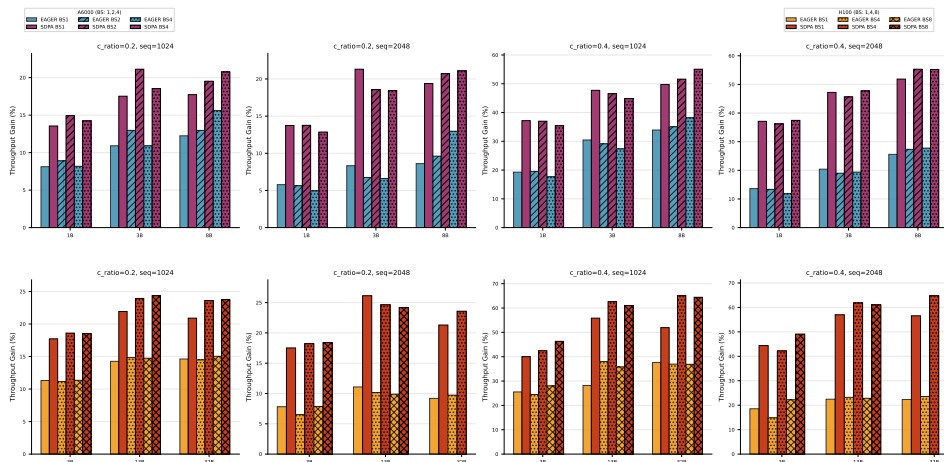

Figure 7: Throughput gains of $A^3$ relative to the uncompressed model across a range of settings. **C_ratio** denotes the compression ratio, and **Seq** denotes the sequence length. The top row shows results on a single A6000, while the bottom row shows results on a single H100.

@Reviewer gBeN Figure 8 shows how offline compression time grows with model size. We evaluate 20% compression on a single H100 GPU for Llama-3.2-1B, Llama-3.2-3B, Llama-3.1-8B, Qwen3-32B, and Llama-3-70B. Overall, compression time increases rapidly with model scale because many operations in $A^3$ have computational complexity greater than $O(n^2)$, where $n$ is the model's hidden dimension

It is worth noting that the compression of each layer is fully independent, so in practice $A^3$ can be parallelized across $n$ GPUs to achieve roughly $n$ times speedup.

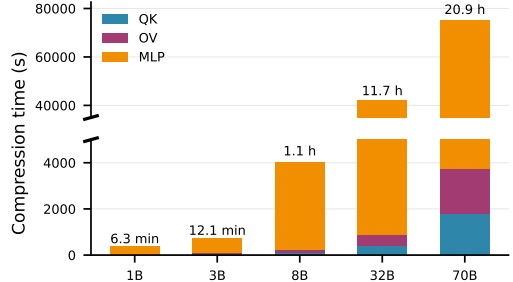

Figure 8: $A^3$ offline compression time breakdown across model sizes

For GQA's QK projections with RoPE, $A^3$-QK becomes increasingly expensive as hidden size grows. By the time the model reaches 70B parameters, $A^3$-QK accounts for more than 90% of total compression time due to the cost of joint decomposition.

## G PERFORMANCE ON PHI MODEL

To evaluate the performance of $A^3$ across other model families, we provide additional results for the Phi models following Table 1 and Table 2 metrics. Thanks to its optimization-based design, $A^3$ continues to perform very well for this model family.

| Compression | Method | Wikitext-2 | SlimPajama | C4 |
|---|---|---|---|---|
| 10% | SVD-LLM | 6.81 (+2.50) | 8.40 (+1.69) | 10.47 (+1.66) |
| | $A^3$ | **5.44 (+1.14)** | **7.28 (+0.58)** | **9.48 (+0.67)** |
| 20% | SVD-LLM | 8.14 (+3.83) | 9.67 (+2.96) | 11.90 (+3.10) |
| | $A^3$ | **6.40 (+2.10)** | **8.16 (+1.46)** | **10.59 (+1.79)** |

Table 13: A comparison of phi-3-medium-4k-instruct (14B) perplexity ($\downarrow$) on WikiText2, C4, and SlimPajama.

| Compression | Method | ARC Challenge | BoolQ | OpenbookQA | GSM8K (Strict) | MMLU | Avg |
|---|---|---|---|---|---|---|---|
| - | Original | 0.6672 | 0.8850 | 0.4120 | 0.8279 | 0.7797 | 0.7144 |
| 10% | SVD-LLM | 0.5751 | 0.8703 | 0.3720 | 0.6179 | 0.7134 | 0.6297 |
| | $A^3$ | 0.6118 | 0.8841 | 0.3800 | 0.7589 | 0.7340 | **0.6738** |
| 20% | SVD-LLM | 0.5034 | 0.8618 | 0.3260 | 0.4913 | 0.6773 | 0.5720 |
| | $A^3$ | 0.5273 | 0.8645 | 0.3480 | 0.6073 | 0.6715 | **0.6037** |

Table 14: A comparison of downstream task accuracy ($\uparrow$) of phi-3-medium-4k-instruct (14B).

