# OpenReview forum: "A3 : an Analytical Low-Rank Approximation Framework for Attention"
_ICLR.cc/2026/Conference — Submitted to ICLR 2026_

### Official Review · Reviewer_5VwV · 2025-10-27

**Soundness:** 2
**Presentation:** 2
**Contribution:** 1
**Rating:** 2
**Confidence:** 5

**Summary:**

This work proposes an analytical method for post-training low rank approximation of transformer language models. Experiments are conducted on LLaMA models of various sizes (7B to 70B parameters) with compression ratios of 10% and 20% for most results (with up to 60%).

**Strengths:**

Making language models compact and efficient is an important topic.

**Weaknesses:**

There are two major weaknesses: (1) the proposed method is highly incremental, and (2) the practical significance of the results is very limited.

* Method: Generally speaking, a post-training low-rank approximation method for transformer language models falls within the realm of minor engineering or optimization tweaks rather than representing a broadly impactful algorithmic contribution. Unless there is significant algorithmic innovation or strong empirical results, such an approach is insufficiently impactful.

* Experimental results: In the end, there is substantial performance degradation even with only a 10% compression ratio compared to the uncompressed model. As a result, while the method might be relevant for some very specific application scenarios, it does not stand out as a broadly useful or competitive approach.

For these reasons, while the paper may be suitable for an engineering-oriented venue focusing on language model compression, it falls well below the standard expected at general machine learning conferences.

**Questions:**

The reviewer has no further questions and considers it unlikely that this work will become acceptable after any rebuttal or discussion, given that the limitation lies in the core idea of the proposed method itself.

---

> ### Author Response · Authors · 2025-11-18
> **Q1-1**
>
> > Method: Generally speaking, a post-training low-rank approximation method for transformer language models falls within the realm of minor engineering or optimization tweaks rather than representing a broadly impactful algorithmic contribution. Unless there is significant algorithmic innovation or strong empirical results, such an approach is insufficiently impactful.
>
> We thank the reviewer for leaving comments and suggestions. We would like to address your concerns one by one. Firstly, we explain why the low-rank approximation method is not minor engineering. Secondly, we highlight our contribution compared to previous works in the domain and show that we have a clear improvement against prior work also at high compression scale.
>
> **Post-training low-rank approximation is a valuable research topic**, instead of “the realm of minor engineering or optimization tweaks”
> - Hardware Independence: Other post-training methods such as quantization typically require specialized hardware support to realize real speedups. Without such support, dequantization introduces computational overhead—memory is saved, but actual throughput does not improve. In contrast, low-rank decomposition methods such as $A^3$ reliably offer **both memory savings and throughput speedup on any hardware**, with no need for special instructions/hardware support.
> - **Finer Control and Extreme Compression:** For extreme compression regimes, combining low-rank methods with quantization creates a **continuous spectrum of compression levels** between discrete quantization points and can yield a way better Pareto frontier. Here we extend the “Quantization compatibility” experiments in Appendix 5. As shown in Fig. W2 sub-3-bit **quantization alone destabilizes the model**, whereas **$A^3$ + Quantization** provides a significantly better accuracy vs compression trade-off. [Figure Q1. Lower perplexity enabled by $A3$ + quantization.](https://bashify.io/i/1PEH4V)
> - **Complementarity with Other Compression Method:** In fact, many recent works, [SVDQuant](https://arxiv.org/abs/2411.05007), [QERA](https://arxiv.org/abs/2410.06040), [Slim](https://arxiv.org/abs/2410.09615), [LQER](https://arxiv.org/abs/2402.02446), [OATS](https://arxiv.org/abs/2409.13652)**, among others, explore how to combine the strengths of both Low-rank and compression techniques such as quantization, pruning to achieve superior overall compression. Investigating how best to integrate $A^3$ with quantization is a promising direction, but it lies outside the scope of this paper.

---

> ### Author Response · Authors · 2025-11-18
> **Q1-2**
>
> Besides our analytical solutions, **we do have strong empirical results.**
> - We used a compression ratio of 10% and 20% in Table 1 and 2 to keep the resultant perplexity sensible, because none of the SoTa methods low-rank methods yield LLMs with usable perplexity for a larger compression ratio like 40%. In Figure 2, we already show that with a compression ratio of 40% and 60%, $A^3$ still outperforms SoTA methods even though a large perplexity like 33 indicates an unusable LLM.
> - In the tables below, we offer more experiments on extreme compression ratios.
>     - At the same compression ratio, [Clover](https://openreview.net/forum?id=Z7ms8FJfCM), [Palu](https://arxiv.org/abs/2407.21118), and $A^3$ reduce parameters and KV-cache size by exactly the same amounts. However, their perplexity behaves very differently. Clover’s perplexity collapses immediately starting at 20% compression, while Palu performs closer to $A^3$ but not better as its objective only reduces K, V projection layer output loss.
>     - $A^3$ outperforms all three methods. As the compression ratio increases, this advantage becomes more pronounced: $A^3$ is the only method that remains below 1k perplexity even at 80% compression.
>
> Table Q1. A comparison of perplexity ($\downarrow$) on WikiText2, C4, and SlimPajama.
> | MPT-7B |        SlimPajama        |               |               |          C4          |               |               |       Wikitext-2       |               |               |
> |--------|---------------------------|---------------|---------------|-----------------------|---------------|---------------|-------------------------|---------------|---------------|
> |CRatio | Clover                    | Palu          | $A^3$           | Clover                | Palu          | $A^3$           | Clover                  | Palu          | $A^3$           |
> | **20%** | 48.11                    | 9.67          | **8.88**          | 53.29                 | 11.74         | **10.77**         | 77.78                   | 8.73          | **8.05**          |
> | **40%** | 383                      | 11.51         | **9.90**          | 408                   | 14.18         | **12.20**         | 795                     | 10.60         | **9.19**          |
> | **60%** | 5397                     | 25.73         | **15.34**         | 4919                  | 32.26         | **18.71**         | 7895                    | 25.09         | **15.58**         |
> | **80%** | 15467                    | 5270          | **388**           | 11661                 | 3210          | **373**           | 14434                   | 13714         | **849**           |
>
>
> | MPT-30B |        SlimPajama        |               |               |          C4          |               |               |       Wikitext-2       |               |               |
> |--------|---------------------------|---------------|---------------|-----------------------|---------------|---------------|-------------------------|---------------|---------------|
> |CRatio | Clover                    | Palu          | $A^3$           | Clover                | Palu          | $A^3$           | Clover                  | Palu          | $A^3$           |
> | **20%** | 11.52                    | 7.91          | **7.71**          | 14.53                 | 9.87          | **9.59**          | 13.07                   | 7.04          | **6.73**          |
> | **40%** | 18.00                    | 8.99          | **8.33**          | 22.43                 | 11.30         | **10.44**         | 23.47                   | 8.40          | **7.40**          |
> | **60%** | 54.97                    | 15.59         | **11.52**         | 70.65                 | 18.91         | **14.22**         | 95.45                   | 18.88         | **11.28**         |
> | **80%** | 779                      | 211           | **37.09**         | 732                   | 253           | **42.85**         | 1524                    | 339           | **46.72**        |
>
> Throughout our papers, we show $A^3$ method outperforms various SoTA methods in terms of perplexity, accuracy, GPU throughput Etc.

---

> ### Author Response · Authors · 2025-11-18
> **Q1-3**
>
> Here we restate our contributions to argue that **we are research-oriented work with novel & grounded math derivations and sufficient empirical experiments to support our claims of outperforming SoTA performance.**
> - $A^3$ formulation offers a better end-to-end modelling than previous works, hence better accuracy: We propose a three part low-rank approximation setup for transformer blocks and derive closed-form solutions for the three objectives.
>     - Different from approximating by minimising each linear layer output error, our formulation is a closer representation to the analytical approximation of the end-to-end model performance.
>     - Our formulation trims both FLOPs/Memory and information energy proportionally to the rank $r$, which offers a more effective trade-off.
>
> This strong formulation allows us to perform a clear margin in terms of accuracy to previous SoTA works in low-rank model compression and KV-Cache compression (Clover, Palu, SVD-LLM).
> - $A^3$ compression is directly translatable to runtimes speed up with no overhead introduced. Unlike SVD-LLM that breaks each linear layer into two matrices, which requires an extra GEMM kernel, $A^3$ reduces the native dimension in the model. This naturally reduces model sizes, KV cache sizes, FLOPs and avoids runtime overheads.
> - We adapted $A^3$ for diverse Transformer architecture, including GQA and RoPE. This overcomes the limitation of clover and palu which can’t achieve actual speed up with RoPE.
>
> $A^3$ presents a novel post-training low-rank decomposition method. It is an off the shelf method that performs better than the SoTA on its own. For more aggressive model compression, we provide an early signal that shows $A^3$ works with fine-tuning and quantization, however extensively exploring these two directions is out of the scope of the current paper.

---

> ### Author Response · Authors · 2025-11-18
> **Q2**
>
> > Experimental results: In the end, there is substantial performance degradation even with only a 10% compression ratio compared to the uncompressed model. As a result, while the method might be relevant for some very specific application scenarios, it does not stand out as a broadly useful or competitive approach.
>
> We do have strong empirical results. Please see our reply to Q1-2.

---

> > ### Comment · Reviewer_5VwV · 2025-11-18
> >
> > I sincerely thank the authors for their response and the updated manuscript.
> >
> > However, the authors’ response does not address any of my main concerns.
> >
> > First, and very critically, I would like to ask the authors to reserve the phrase "state of the art" for truly state-of-the-art methods. We quickly risk inflation of this term once it is applied too broadly---for example, "SoTA low-rank–based compression methods for transformer language models", "SoTA quantization-based compression methods," "SoTA pruning," "SoTA distillation," and so on.
> >
> > The goal should be to obtain the best compression method across these variations, and the best across categories is what constitutes the true state of the art.
> > Here, the comparison is limited to low-rank–based methods only (SVD, Clover, Palu); these results alone cannot represent any "SoTA" in its true sense.
> >
> > > Post-training low-rank approximation is a valuable research topic, instead of "the realm of minor engineering or optimization tweaks"
> >
> > Minor engineering does not imply that a topic is not valuable. However, this is ICLR---a general machine learning conference that goes beyond language-focused venues such as *ACL, EMNLP, or COLM.
> > From that standpoint, the submission is primarily application-focused and engineering-oriented, and it does not provide compelling algorithmic originality or strong empirical results.
> >
> > > we do have strong empirical results.
> >
> > It is not sufficient to simply claim that the results are "strong". As I already commented in my original review, any compression with a rate higher than 20% causes excessive performance degradation here, making the resulting models impractical.
> > Therefore, discussing or comparing results obtained at such high compression rates is of limited practical relevance; therefore the impact of the new Table 3 is limited: all the models perform poorly in this regime, some worse than others but all unsatisfactorily.
> >
> > For these reasons, I will maintain my current score of 2.

---

> ### Author Response · Authors · 2025-11-18
>
> Dear Reviewer 5VwV,
>
> Thank you for the comments. We appreciate the opportunity to engage in a constructive discussion, as it allows us to learn from each other’s perspective and understand differing perspectives on novelty and contribution in this research area.
>
> From our perspective, the evaluation of a paper should focus on **its technical contributions rather than on comparisons shaped by methodological/conceptual preferences or viewing the field as an engineering competition**. We even have researchers working on classic ML algorithms, like deep forest, we can't disregard them simply because DL-based methods outperform them. As a community, we need to encourage all fields in the domain to make progress rather than us overfitting to a single direction.
>
> In our submission, we provided a detailed description of our novel algorithmic framework. While low-rank approximation is an established problem setting, **the formulation presented in Section 3 and its closed-form solution constitute new contributions**. If these are viewed as incremental or previously known, we would greatly appreciate pointers to prior mathematical formulations or proofs that resemble our approach.
>
> Regarding the comment that our results are “not strong,” we would like to reiterate that our method achieves best-in-class performance within the domain of low-rank approximation [1,2,3]. Additionally, we show that **our method works well with quantization**, an area you highlighted, providing an additional 20% improvement and outperforming quantization-only methods in extreme compression settings. We kindly invite you to revisit the results in Section 4, Appendix D, and Appendix E, which detail strong compression rates, compatibility with quantization, fine-tuning behavior, and runtime performance.
>
> Continuing from the performance point of view, our updated experiments show compression rates scaling from 20% to 80% for MPT-7B and MPT-30B, we achieved an order of magnitude higher than previous approaches [2,3]. $A^3$ reveals that for MHA with no RoPE, QK and OV components are highly low-rank. The model is still functional when dropping 50% of the head dimension away (**50% of the parameter counts**). We believe this insight is particularly valuable not only to the compression field but also to model architecture design and mechanistic interpretability on attention mechanisms. We also point out that CUR decomposition is the core reason that performance deviates from the analytical solution at high compression rates. Future directions in low-rank could investigate new mathematical formulation to improve on this. We believe these are also critical values that a research paper should ship – pointing out caveats and areas for later improvements that a field can make.
>
> Thank you again for your time and reply. We hope these clarifications help convey the technical significance and broader value of our contributions. We look forward to hearing your further feedback. We remain open to continue discussions on any aspect of the paper.
>
> - [1] SVD-LLM: Truncation-aware singular value decomposition for large language model compression. ICLR 2025
> - [2] Palu: Compressing kv-cache with low-rank projection. ICLR 2025
> - [3] CLOVER: Cross-Layer Orthogonal Vectors Pruning. ICML 2025

---

> > ### Comment · Reviewer_5VwV · 2025-11-18
> >
> > Once again, I thank the authors for their response.
> >
> > > We even have researchers working on classic ML algorithms, like deep forest, we can't disregard them simply because DL-based methods outperform them. As a community, we need to encourage all fields in the domain to make progress rather than us overfitting to a single direction.
> >
> > Research in random forests or non-neural count-based language modeling is certainly welcome. However, as a reviewer, my responsibility is to evaluate submissions critically and selectively. The key question I must address is the significance and impact of the proposed method within the current research context.
> >
> > My evaluation is not based on the research topic itself, but on the content and contribution of the paper. I do refer to the topic and the nature of the paper because different topics naturally carry different expectations in terms of technical depth, novelty, and empirical demonstration.
> >
> > There are some fundamental questions to consider: Would this method be practically adopted for model compression at 20% and beyond? Based on the presented results, my answer would be no. Therefore, I find the proposed method to be insufficiently impactful in practice. Referring to the approach as "SoTA" within the limited domain does not materially change this assessment (as I emphasized in my response, such usage of "SoTA" is misleading).
> >
> > Another question is whether the work provides significant algorithmic insights that outweigh its limited empirical success. Here again, my answer is negative. While the presence of closed-form solutions is appreciated, it alone does not constitute  impactful contributions or insights---There are good reasons why venues such as TMLR (which focuses more on correctness) or ACL/EMNLP and now COLM (which emphasize language applications) may be more suitable for this line of work. I am aware that there are antecedents, but in my view, the overall impact of this submission falls below the standard that ICLR should aim to uphold.
> >
> > I therefore maintain my current score but will ultimately defer to the AC's judgment.

---

> > > ### Author Response · Authors · 2025-12-01
> > >
> > > Dear Reviewer 5VwV,
> > > Thank you once again for your thoughtful comments. Regarding your specific question — *"Would this method be practically adopted for model compression at 20% and beyond?"* — We would like to provide additional clarification and supporting results.
> > > First, we reiterate that **$A^3$ achieves state-of-the-art performance within the low-rank approximation domain** [1,2,3] as mentioned in Q1. With the addition of light fine-tuning, same as SVD-LLM [1], we show that **$A^3$ can recover back to a fully usable model (ΔPPL < 1) while providing up to additional 40% compression on top of quantization on Attention FLOPs/Parameter/KV Cache on MPT-30B**, thanks to its strong initialization of ranks in each component. Additionally, for extreme compression, $A^3$ combined with Quantization can offer a better trade-off compared to quantization alone (4-bit HQQ + $A^3$ @ 60% vs Pure 2-bit HQQ Quantization).
> > > ### Table R1: MPT-30B ΔPPL relative to baseline (no compression) on WikiText2; lower is better
> > > | Method | Attention FLOPs/Parameter/KV Cache Compression Ratio | Without fine-tuning | With fine-tuning |
> > > | --------------------------------------------- | ------ | ----- | ----- |
> > > | Dense | 1x | 0 | 0 |
> > > | Pure 4-bit HQQ Quantization | 4x | +0.11 | - |
> > > | Pure 2-bit HQQ Quantization | 8x | +12.78 | +2.80 |
> > > | 4-bit HQQ + $A^3$ @ 20% | 5x | +0.99 | +0.59 |
> > > | 4-bit HQQ + $A^3$ @ 40% | 6.67x | +1.15 | +0.99 |
> > > | 4-bit HQQ + $A^3$ @ 60% | 10x| +18.15 | +2.74 |
> > > Finally, as we discussed in Appendix D, due to the limitation of CUR decomposition, if the user wants to achieve an extreme compression ratio in Parameter Count, a good practice will be a proper combination of $A^3$ and SVD-LLM based on the model's given architecture. E.g ($A^3$-QK, $A^3$-VO, SVD-LLM for MLP). Exploring the optimal combination is out of the scope of this paper.
> > > Thank you again for your time and question. We hope these clarifications and results fully address your question regarding the practical adoption of $A^3$ above 20%.

---

### Official Review · Reviewer_voSr · 2025-10-28

**Soundness:** 2
**Presentation:** 3
**Contribution:** 2
**Rating:** 4
**Confidence:** 4

**Summary:**

A3 is an analytical low-rank approximation (LRA) framework that operates at the functional component level of Transformer layers—QK, OV, and MLP—instead of factorizing individual weight matrices. It provides closed-form reductions for QK and OV, and a CUR-style selection for MLP, with adaptations for GQA and RoPE. Claimed benefits: lower PPL at the same compression ratio, reduced KV-cache and FLOPs, and end-to-end speedups without extra kernel launches.

**Strengths:**

- **Component-aware objectives.** Clear decomposition into QK/OV/MLP with matched optimization goals; closed-form A3-QK (Thm. 2) and A3-OV (Thm. 3) are principled and easy to implement.
- **No extra GEMMs.** Unlike matrix factorization $W \approx AB$ that adds a matmul, A3 shrinks head dimensions and FFN width, preserving operator count while directly cutting FLOPs and KV-cache. End-to-end speedups are reported (e.g., Fig. 3).
- **Covers GQA and RoPE.** Joint-SVD for GQA and CUR over RoPE frequencies extend applicability beyond vanilla MHA.

**Weaknesses:**

- **Omission of SVD-LLM v2 in main tables.** The text lists SVD-LLM v2 as a baseline, but Tables 1–2 compare only to SVD-LLM. Please add head-to-head v2 results in the main tables or justify exclusion.
- **Limited compression-ratio coverage.** Core tables report only 10% and 20%. Provide 30–60% for all models to show scaling behavior, not just a single model/figure.
- **Invertibility assumptions.** Theoretical parts assume invertible/positive-definite correlation matrices $R$. Clarify when $R$ can be singular or ill-conditioned in practice (e.g., sample-poor regimes), what regularization is used (SVD pseudo-inverse, ridge, shrinkage), and how this affects accuracy.

**Questions:**

- **SVD-LLM v2.** You list v2 as a baseline. Why is it absent from Tables 1–2? Please provide direct v2 numbers under the same three-component setting or explain incompatibilities.
- **Separate $W_Q$ and $W_K$ in practice.** Eqs. (9)–(13) conceptually fuse $W_Q$ and $W_K$ during analysis, then yield reduced-dimensional projections. In real stacks that use split $W_Q/W_K$, how do you deploy the reduced head dimension without breaking fused-QK kernels, KV-cache layout, or tensor-parallel shards?

---

> ### Author Response · Authors · 2025-11-17
> **Q1**
>
> Thank you for the valuable suggestions. We would like to answer your questions one by one. Meanwhile, the revised parts of the manuscript are colored in blue and the changes reflecting your comments are highlighted with **"@Reviewer voSr"** in purple. The line numbers are based on the revised version.
>
> > Omission of SVD-LLM v2 in main tables. The text lists SVD-LLM v2 as a baseline, but Tables 1–2 compare only to SVD-LLM. Please add head-to-head v2 results in the main tables or justify exclusion.
>
> Thank you for the question. We'd like to provide some clarification regarding SVD-LLM-v2.
>
> SVD-LLM-v2 proposes **a rank allocation algorithm** based on the truncation loss of each layer. However, the truncation loss across different layers are not comparable, and the author did not release a working implementation (See the issues in [their repo](https://github.com/AIoT-MLSys-Lab/SVD-LLM/issues)). We invested considerable time reproducing the SVD-LLM v2 algorithm based on their paper and made sure it reproduces the truncation loss graph (Figure 3 in the [SVD-LLM-v2 paper](https://arxiv.org/pdf/2503.12340)), however we were not able to replicate the end-to-end model performance results reported in the main table.
>
>
> **Reproducibility issues with SVD-LLM-v2 have also been noted by other users in the SVD-LLM GitHub repository, where several open issues [[issue 1](https://github.com/AIoT-MLSys-Lab/SVD-LLM/issues/35),[issue 2](https://github.com/AIoT-MLSys-Lab/SVD-LLM/issues/47),[issue 3](https://github.com/AIoT-MLSys-Lab/SVD-LLM/issues/46)] highlight problems with rank assignment and inconsistent behavior across models.** In our testing, we evaluated the algorithm on multiple models including MPT-7B, LLaMA-2-7B, LLaMA-2-13B, and LLaMA-3.1-8B. We were able to generate results for Meta-LLaMA-3.1-8B that match the qualitative trends of Figure 3 in SVD-LLM-v2 paper. However, for other models, the algorithm often either produced invalid (e.g., negative) rank assignments or failed to assign ranks altogether due to either extremely small or large truncation losses.
>
> We also contacted the original authors, who confirmed that the rank allocation algorithm is highly sensitive to the choice of the calibration set. This sensitivity likely contributes to the inconsistencies in reproducibility across different setups.
>
> Below are the results we obtained for LLaMA-3.1-8B:
> | Compression | Method     | ARC Challenge | BoolQ  | OpenbookQA | Winogrande | GSM8K (Strict) | MMLU   | **Avg**    |
> | ----------- | ---------- | ------------- | ------ | ---------- | ---------- | -------------- | ------ | ---------- |
> | -           | Original   | 0.5401        | 0.8190 | 0.3340     | 0.7822     | 0.4920         | 0.6535 | 0.6035 |
> | 10%         | SVD-LLM    | 0.3575        | 0.7458 | 0.2600     | **0.7111**     | 0.0447         | 0.4708 | 0.4317 |
> |             | SVD-LLM-v2 | 0.3566        | 0.7504 | 0.2540     | 0.7103     | 0.0523         | 0.4672 | 0.4318 |
> |             | $A^3$      | **0.4565**   | **0.7884** |**0.3180**     | 0.7072     |**0.2388**         | **0.5922** | **0.5168** |
> | 20%         | SVD-LLM    | 0.2534        | **0.6948** | 0.2200     | **0.6440**     | 0.0113         | 0.3604 | 0.3640 |
> |             | SVD-LLM-v2 | 0.2397        | 0.6920 | 0.2320     | 0.6377     | 0.0113         | 0.3476 | 0.3601 |
> |             |  $A^3$     | **0.3345**   | 0.6823 | **0.2520**     | 0.6417     |**0.0705**         | **0.4649** | **0.4076** |
>
> Additionally, we would like to clarify that SVD-LLM v2 is a rank-allocation method: it searches for a good rank assignment for each layer given a compression budget, **but their low-rank approximation algorithm remains the same as SVD-LLM v1.** In contrast, our experiments in $A^3$ use a uniform rank across the entire model. Exploring rank-allocation strategies within $A^3$ is a promising direction for achieving additional performance gains, but extensive investigation lies beyond the scope of the current paper.

---

> ### Author Response · Authors · 2025-11-18
> **Q2**
>
> > Limited compression-ratio coverage. Core tables report only 10% and 20%. Provide 30–60% for all models to show scaling behavior, not just a single model/figure.
>
> Thank you for the question. We didn’t provide every compression ratio from 10% to 60% for  each model in core tables due to two reasons:
> - Limited resources. We spent around 2000 GPU hours to collect the existing results in the paper. Running all {compression ratio, model, task} combinations is beyond the capability of our research group.
> - For SoTA post-training low-rank methods, a compression ratio larger **>= 40%** usually results in **an unusable model**. For example, in Figure 2, a llama-7b with c_ratio=40% yields a perplexity > 20. In this case, even $A^3$ still clearly outperforms the baselines, a large perplexity like 33 indicates an unusable LLM. For example, tiny-llama 1.1b achieves perplexity around 14 with only 1/7 parameters of llama-7b. Thus we stop the compression at 20% to ensure a sensible evaluation.
>
> Here we provide additional $A^3$-QK/VO experiments and show the scaling behaviour on two models (MPT-7b, MPT-30b) across compression ratio 20% , 40%, 60%, and 80%.
>
>
> Table Q2: A comparison of perplexity ($\downarrow$) on WikiText2, C4, and SlimPajama on MPT-7B and MPT-30B.
> | MPT-7B |        SlimPajama        |               |               |          C4          |               |               |       Wikitext-2       |               |               |
> |--------|---------------------------|---------------|---------------|-----------------------|---------------|---------------|-------------------------|---------------|---------------|
> |CRatio | Clover                    | Palu          | $A^3$           | Clover                | Palu          | $A^3$           | Clover                  | Palu          | $A^3$           |
> | **20%** | 48.11                    | 9.67          | **8.88**          | 53.29                 | 11.74         | **10.77**         | 77.78                   | 8.73          | **8.05**          |
> | **40%** | 383                      | 11.51         | **9.90**          | 408                   | 14.18         | **12.20**         | 795                     | 10.60         | **9.19**          |
> | **60%** | 5397                     | 25.73         | **15.34**         | 4919                  | 32.26         | **18.71**         | 7895                    | 25.09         | **15.58**         |
> | **80%** | 15467                    | 5270          | **388**           | 11661                 | 3210          | **373**           | 14434                   | 13714         | **849**           |
>
>
> | MPT-30B |        SlimPajama        |               |               |          C4          |               |               |       Wikitext-2       |               |               |
> |--------|---------------------------|---------------|---------------|-----------------------|---------------|---------------|-------------------------|---------------|---------------|
> |CRatio | Clover                    | Palu          | $A^3$           | Clover                | Palu          | $A^3$           | Clover                  | Palu          | $A^3$           |
> | **20%** | 11.52                    | 7.91          | **7.71**          | 14.53                 | 9.87          | **9.59**          | 13.07                   | 7.04          | **6.73**          |
> | **40%** | 18.00                    | 8.99          | **8.33**          | 22.43                 | 11.30         | **10.44**         | 23.47                   | 8.40          | **7.40**          |
> | **60%** | 54.97                    | 15.59         | **11.52**         | 70.65                 | 18.91         | **14.22**         | 95.45                   | 18.88         | **11.28**         |
> | **80%** | 779                      | 211           | **37.09**         | 732                   | 253           | **42.85**         | 1524                    | 339           | **46.72**        |
>
>
> Here are the figure version of the table:
> - [$A^3$, Clover, Palu scale performance on MPT-7b](https://bashify.io/i/gcVE2Q)
> - [$A^3$, Clover, Palu scale performance on MPT-30b](https://bashify.io/i/0DL1yX)
>
> We show that the benefits of $A^3$-QK and $A^3$-OV widens with compression ratios and model size, enabling superior performance in Attention FLOPs/Parameter Counts compression and KV Cache compression. For example, in MPT-30B table, **at c_ratio=80%**, Palu yields a perplexity of 339 on WikiText2, while $A^3$ is 46.72, though none of these perplexity indicates an usable LLM considering the model size is 30B.
>
> Moreover, as we discussed in the Appendix D. Due to the limitation of CUR decomposition, If the user wants to achieve an extreme compression ratio, a good practice will be a proper combination of $A^3$ and SVD-LLM based on the model given architecture. E.g ($A^3$-QK, $A^3$-VO, SVD-LLM for MLP). Exploring the optimal combination is out of the scope of this paper.

---

> ### Author Response · Authors · 2025-11-18
> **Q3**
>
> > Invertibility assumptions. Theoretical parts assume invertible/positive-definite correlation matrices . Clarify when  can be singular or ill-conditioned in practice (e.g., sample-poor regimes), what regularization is used (SVD pseudo-inverse, ridge, shrinkage), and how this affects accuracy.
>
> This symmetric semi-positive definite auto-correlation matrix is a well known matrix used in many recent works like GPTQ, SVD-LLM, QERA, etc. Following the [implementation in GPTQ](https://github.com/IST-DASLab/gptq/blob/2d65066eeb06a5c9ff5184d8cebdf33662c67faf/gptq.py#L98), we add damping to zero eigenvalues of the autocorrelation matrix. Additionally, we found in all of our experiments across different calibration datasets, the autocorrelation matrices were always invertible. We have added this in the Appendix C Approximation paragraph in the revised manuscript.

---

> ### Author Response · Authors · 2025-11-18
> **Q4**
>
> > SVD-LLM v2. You list v2 as a baseline. Why is it absent from Tables 1–2? Please provide direct v2 numbers under the same three-component setting or explain incompatibilities.
>
> Please refer to our reply to Q1.

---

> ### Author Response · Authors · 2025-11-18
> **Q5**
>
> > Separate  and  in practice. Eqs. (9)–(13) conceptually fuse Wk and Wq during analysis, then yield reduced-dimensional projections. In real stacks that use split Wq/Wk, how do you deploy the reduced head dimension without breaking fused-QK kernels, KV-cache layout, or tensor-parallel shards?
>
>
> Thanks for the question. The short answer is our A3 is a plug-and-play method that reduces the problem size without breaking any existing kernels.
>
> Take SPDA as an example which uses the fused kernel, it takes queries and keys as input, where $Q = X_q W_q$ and $K = X_{kv} W_{kv}$. What $A^3$ does is compress the product $W_q W_{kv}^T$ into two smaller matrices: $\tilde{W}_q$ and $\tilde{W}_{kv}^T$. **The shape of key/query weight matrix is changed from $[\mathrm{hidden\_dim}, \mathrm{head\_dim}]$ to $[\mathrm{hidden\_dim}, r]$**, where $r \ll \mathrm{head\_dim}$. These act as the new projection matrices for query $Q$ and key $K$ generation.
>
> As a result, the input projections still follow the form $Q = X_q \tilde{W}_q$ and $K = X_{kv} \tilde{W_{kv}}$, but with a reduced head dimension — **the shape of key/query matrix is changed from $[\mathrm{seq\_len}, \mathrm{head\_dim}]$ to $[\mathrm{seq\_len}, r]$**, where $r \ll \mathrm{head\_dim}$. This compressed $Q$ and $K$ are then processed by SDPA. **Hence, no custom kernel is required. In practice, to further optimize performance after $A^3$ compression, users may use auto-tuning to find the optimal kernel block size for the new problem size**, but there is no need to change the kernels or KV cache management.
>
>
> In **Appendix.E**, we present a comprehensive runtime analysis of $A^3$, covering throughput and peak memory usage with and without the SPDA attention implementation. These results show that the $A^3$ compression ratio translates directly into runtime improvements that closely match theoretical expectations, all without requiring changes to existing kernels or memory management.
>
>
> Here we present more evidence that demonstrates the throughput gains of $A^3$ are robust across different GPUs, batch sizes, compression ratios, and sequence lengths with and without SDPA fused kernels.
>
> - [Figure Q5: More throughput evidence](https://bashify.io/i/HhVA0K)
>
> As expected, $A^3$ consistently delivers strong and stable throughput gains.

---

> > ### Comment · Reviewer_voSr · 2025-11-18
> >
> > The author has addressed most of my concerns, so I am increasing my score to 6.

---

### Official Review · Reviewer_n81d · 2025-11-01

**Soundness:** 3
**Presentation:** 3
**Contribution:** 3
**Rating:** 6
**Confidence:** 3

**Summary:**

The paper proposes A3, a post‑training low‑rank approximation framework tailored to Transformer functional components rather than individual linear layers. A3 decomposes each layer into QK (queries-keys), OV (values-output), and MLP, and derives analytical solutions that reduce the shared hidden dimensions within each component: head dims for QK/OV and the MLP intermediate size. For MHA without RoPE, A3-QK and A3-OV have closed-form weighted SVD solutions that minimize pre‑softmax attention score error and per‑head attention output error, respectively. For MLP and for attention with RoPE, A3 uses CUR-style selection heuristics to identify important columns/rows (and RoPE frequency pairs). The method also adapts to GQA via joint SVD across grouped heads. Because it shrinks native dimensions rather than decomposing layers into two factors, A3 reduces FLOPs, KV cache, and parameters without adding runtime kernels. Experiments on LLaMA-2/3.1 (7B–70B), MPT, and Phi show large perplexity and downstream accuracy gains over SVD-LLM and other low-rank baselines at matched compression budgets, along with consistent throughput improvements.

**Strengths:**

- **Clear component-wise reformulation** (QK, OV, MLP) that aligns local objectives with Transformer functionality, leading to head/intermediate dimension reduction rather than two-factor decompositions.

- **Closed-form weighted SVD solutions** for QK/OV (MHA-NoPE) with principled use of activation autocorrelations; practical GQA joint-SVD extension.

- **Efficiency**: Avoids extra GEMM kernels and reduces KV cache and attention FLOPs; verified throughput gains on GPU.

- **Strong empirical results** across multiple model families and sizes (7B–70B) and tasks, often with large margins over SVD-LLM at 10–20% compression; especially notable PPL improvements on LLaMA-3.1-70B.

- **Sensible ablations** (component-wise, RoPE vs pruning baselines, simplified variants), and demonstrations of compatibility with quantization and mixed-rank allocation.

**Weaknesses:**

**Novelty & positioning**
The key step beyond prior activation-aware low-rank methods (e.g., Eq. 8-style weighted SVD) seems to be the component-wise framing; A3-QK feels close to CLOVER but adds activation whitening. To make the novelty pop, I’d love to see fuller, apples-to-apples comparisons to CLOVER and Palu at scale (not just small ablations), plus a short what’s truly new vs. reinterpreted paragraph in the intro.

**Fairness of compression ratio.**
It’s hard to tell whether reduction budget is matched across compute, memory, and KV cache for each baseline. Please spell out the budget definition and how it maps to SVD-LLM and others (e.g., FLOPs saved, KV footprint, kernel count) so the reader can judge fairness.

**Throughput evidence.**
Throughput is reported for a single model/hardware/batch. Since kernel efficiency can swing with head-dim choices (alignment, vectorization, fused kernels), a brief sensitivity sweep (head dim × batch × sequence length) or a second hardware config would improve support.

**Theory–practice gap.**
The QK objective drops the softmax nonlinearity and OV heads are treated independently; that can decouple local error from global attention error. Even a small diagnostic (e.g., correlation between local objective reduction and end-to-end perplexity) or a bound/intuition section would help reconcile this.

**Questions:**

- How exactly is the "compression ratio" defined and matched across methods? Do you equalize total FLOPs (including attention), KV cache, and parameters, or parameter count alone?

- For RoPE, how robust is the CUR frequency-pair selection across datasets and sequence lengths? Any analyses showing which frequencies are retained/dropped and their impact on long-context performance?

- What regularization/pseudoinverse strategy is used when autocorrelation matrices are ill-conditioned? Sensitivity to calibration set size/domain?

- Are you able to run more throughput experiments, e.g., with different batch sizes, sequence lengths, attention backends, and head dimensions (including hardware-friendly multiples)?

---

> ### Author Response · Authors · 2025-11-17
> **Q1-1**
>
> Thank you for careful reviewing and raising questions about experiment setup and results. We would like to answer them one by one. Meanwhile, the revised parts of the manuscript are colored in blue and the changes reflecting your comments are highlighted with "**@Reviewer n81d**" in orange. The line numbers are based on the revised version.
>
> > Novelty & positioning The key step beyond prior activation-aware low-rank methods (e.g., Eq. 8-style weighted SVD) seems to be the component-wise framing; A3-QK feels close to CLOVER but adds activation whitening. To make the novelty pop, I’d love to see fuller, apples-to-apples comparisons to CLOVER and Palu at scale (not just small ablations), plus a short what’s truly new vs. reinterpreted paragraph in the intro.
>
> Thank you for the question. Yes, we would like to provide additional experiments to show the advantage of $A^3$ compared with Clover and Palu at scale. First, we summarize our key contributions and highlight the advantages of $A^3$ relative to Clover and Palu.
>
> ## **1. Stronger Problem Formulation**
>
> A central observation motivating our work is that transformer blocks contain **subcomponents that function as coherent units** (e.g., QK, VO, MLP), each composed of multiple linear layers. Minimizing the **output error of these components jointly** is significantly more effective than minimizing the output error of **each linear layer independently.**
>
> $A^3$ provides both the **formulation** and **the closed-form solutions** for this multi-layer output-error minimization.
> - **Clover** instead minimizes **weight approximation error** for QK and VO. This is equivalent to applying $A^3$ with the autocorrelation matrix fixed to identity, which removes activation information from the objective.
> - **Palu** minimizes the output error of **only the K and V layers**, ignoring contributions from Q and O. We find that these objectives do not adequately model the behavior of the attention mechanism.
>
> In contrast, **$A^3$ explicitly minimizes the output error of both the QK and VO components**, incorporating activation statistics for Q, K, V and the post softmax attention score. This formulation is one step closer to modeling the **true end-to-end behavior** of the transformer block, giving A^3 a strong structural advantage.

---

> ### Author Response · Authors · 2025-11-17
> **Q1-2**
>
> ## **2. Accuracy Advantages Under the Same Compression Ratio**
> Table Q1: A comparison of perplexity ($\downarrow$) on WikiText2, C4, and SlimPajama on MPT-7B and MPT-30B.
> | MPT-7B |        SlimPajama        |               |               |          C4          |               |               |       Wikitext-2       |               |               |
> |--------|---------------------------|---------------|---------------|-----------------------|---------------|---------------|-------------------------|---------------|---------------|
> |CRatio | Clover                    | Palu          | $A^3$           | Clover                | Palu          | $A^3$           | Clover                  | Palu          | $A^3$           |
> | **20%** | 48.11                    | 9.67          | **8.88**          | 53.29                 | 11.74         | **10.77**         | 77.78                   | 8.73          | **8.05**          |
> | **40%** | 383                      | 11.51         | **9.90**          | 408                   | 14.18         | **12.20**         | 795                     | 10.60         | **9.19**          |
> | **60%** | 5397                     | 25.73         | **15.34**         | 4919                  | 32.26         | **18.71**         | 7895                    | 25.09         | **15.58**         |
> | **80%** | 15467                    | 5270          | **388**           | 11661                 | 3210          | **373**           | 14434                   | 13714         | **849**           |
>
>
> | MPT-30B |        SlimPajama        |               |               |          C4          |               |               |       Wikitext-2       |               |               |
> |--------|---------------------------|---------------|---------------|-----------------------|---------------|---------------|-------------------------|---------------|---------------|
> |CRatio | Clover                    | Palu          | $A^3$           | Clover                | Palu          | $A^3$           | Clover                  | Palu          | $A^3$           |
> | **20%** | 11.52                    | 7.91          | **7.71**          | 14.53                 | 9.87          | **9.59**          | 13.07                   | 7.04          | **6.73**          |
> | **40%** | 18.00                    | 8.99          | **8.33**          | 22.43                 | 11.30         | **10.44**         | 23.47                   | 8.40          | **7.40**          |
> | **60%** | 54.97                    | 15.59         | **11.52**         | 70.65                 | 18.91         | **14.22**         | 95.45                   | 18.88         | **11.28**         |
> | **80%** | 779                      | 211           | **37.09**         | 732                   | 253           | **42.85**         | 1524                    | 339           | **46.72**        |
>
> Here are the figure version of the table:
> - [$A^3$, Clover, Palu scale performance on MPT-7b](https://bashify.io/i/gcVE2Q)
> - [$A^3$, Clover, Palu scale performance on MPT-30b](https://bashify.io/i/0DL1yX)
>
>
> All three methods, Clover, Palu, and $A^3$ reduce parameters and KV-cache size **by exactly the same amount** at a fixed compression ratio. However, their **accuracy behaviors differ dramatically**:
> - **Clover** collapses at moderate compression ratios (e.g., perplexity diverges at 20% compression).
> - **Palu** performs more stably but remains worse than $A^3$, as its objective accounts only for K and V output loss.
> - $A^3$ consistently outperforms both:
>     - It maintains significantly lower perplexity across compression ratios, model size and datasets.
>     - This performance gap widens at higher compression levels and with larger model sizes: on MPT-30B, A3 is the only method that stays below 50 perplexity at 80\% compression, and on MPT-7B it is the only one that remains below 1k perplexity.
>
> We have updated this explanation and discussion in the experiment session of the revised manuscript.

---

> ### Author Response · Authors · 2025-11-17
> **Q1-3**
>
> ## **3. Full-Model Post-Training Compression (Attention + MLP)**
> While:
> - Clover primarily targets PEFT-style low-rank adaptation,
> - Palu targets post-training KV-cache compression,
>
> **$A^3$ targets full model compression in a post-training setting.**
>
> We provide formulations for: QK component, VO component, 3-layer MLP block. These three formulations together allow $A^3$ to **reduce parameters**, **shrink KV-cache**, and **provide directly translatable runtime speedups**, without requiring specialized hardware.
> As a result, $A^3$ achieves superior:
> - KV-cache compression (vs. Clover and Palu)
> - Model compression accuracy and throughput gains (vs. SVD-LLM)
>
> We also offer extensions to support modern Transformer architectures like GQA and RoPE.

---

> ### Author Response · Authors · 2025-11-17
> **Q2**
>
> > Fairness of compression ratio. It’s hard to tell whether reduction budget is matched across compute, memory, and KV cache for each baseline. Please spell out the budget definition and how it maps to SVD-LLM and others (e.g., FLOPs saved, KV footprint, kernel count) so the reader can judge fairness.
>
> We define the compression ratio as the **reduced parameter count/total parameter count**. To ensure a fair comparison, we first **match the total number of parameters** of $A^3$ and SVD-LLM.
>
> A transformer layer contains $2 d_m d_h$ parameters for QK, another $2 d_m d_h$ for OV, and $3 d_i d_m$ for the MLP. In $A^3$, the parameter count scales directly with rank, so setting the ranks to $r' = 0.8 d_h$ for QK, OV component and $r' = 0.8 d_i$ for the MLP reduces the number of parameters by 20%. In contrast, SVD-LLM does not have an equal relationship between rank and parameter count, because truncating a matrix from ($d_{\text{in}}, d_{\text{out}}$) to rank $r$ yields $(d_{\text{in}} + d_{\text{out}}) r$ parameters. To achieve the same 20% compression, SVD-LLM use ranks that depend on each layer’s dimensions: for Q, K, V, and O we set $r' = 0.8 \cdot \frac{d_m d_h}{d_m + d_h}$, and for the MLP’s up, gate, and down projections we set $r' = 0.8 \cdot \frac{d_m d_i}{d_m + d_i}$. This ensures that both methods reduce each layer to the same overall parameter count.
>
> Based on this setup, we perform all subsequent measurements, including **FLOPs, KV-cache size, runtime metrics(throughput and peak memory)**, and **perplexity/accuracy**. We have updated the experimental section in the revised manuscript to clarify this setup, and we will open-source our implementation upon acceptance.

---

> ### Author Response · Authors · 2025-11-17
> **Q3**
>
> >Throughput evidence. Throughput is reported for a single model/hardware/batch. Since kernel efficiency can swing with head-dim choices (alignment, vectorization, fused kernels), a brief sensitivity sweep (head dim × batch × sequence length) or a second hardware config would improve support.
>
>
> Yes, we would like to extend the throughput analysis section and provide additional throughput results spanning **GPU type, batch size, model size, compression ratio, sequence length, and attention kernel implementation**. Specifically, we offer GPU throughput results of the following combinations:
>
> - **GPU \& Model Size:** Single A6000 (Llama-3.2-1B/Llama-3.2-3B/Llama-3.1-8B), Single H100 (Llama-3.2-3B/Llama-2-13B/Qwen3-32B)
> - **Batch Size:** 1, 2, 4 for A6000; 1, 4, 8 for H100
> - **Compression Ratio:** 20\%, 40\%
> - **Sequence Length:** 1024, 2048
> - **Attention Kernel:** eager, SDPA
>
> The experiment results can be found here: [Figure.Q3 More throughput evidence](https://bashify.io/i/HhVA0K) (SPDA, seq_len=2048, BS=8 raises OOM on H100 96 GB)
>
> As expected, **$A^3$ consistently provides speedups**, because it reduces the effective problem size **without introducing overhead** from extra kernel launches or underutilization, unlike SVD-LLM. **Larger models/SDPA** benefit more from $A^3$, because besides the linear layer FLOPs, the reduced head dimension saves an extra amount of attention FLOPs that is proportional to ${d_{h}}^2$.
>
> We have created a new subsection in **Appendix E.2** to include these experiments and the accompanying discussion in the revised manuscript.

---

> ### Author Response · Authors · 2025-11-17
> **Q4**
>
> >Theory–practice gap. The QK objective drops the softmax nonlinearity and OV heads are treated independently; that can decouple local error from global attention error. Even a small diagnostic (e.g., correlation between local objective reduction and end-to-end perplexity) or a bound/intuition section would help reconcile this.
>
> Yes, we would like to provide a diagnostic comparing **local objective reductions** to the **end-to-end perplexity**. Before that, we clarify the formulations for **QK** and **OV** in $A^3$.
> - **QK Compression:** Yes, we minimize the attention scores **before the softmax**. This formulation makes the objective linear and allows us to derive a **closed-form solution** for $A^3$-QK which would otherwise be very complicated to solve.
> - **OV Compression:** The post-softmax attention scores are used to perform reductions in OV. In the Appendix B.2.3, we also present a formulation for **jointly compressing all OV heads together** without treating them separately. Though this approach yields a better Pareto frontier in terms of parameter count vs. perplexity, it requires **significantly more KV-cache storage** than even the uncompressed model. From a practical perspective, treating each head independently offers greater overall benefits, particularly for **long-context scenarios.**
>
> ### **Diagnostic Results**
> We compare the effect of **local compression** on QK and OV to the **global end-to-end perplexity:**
> - For **small compression ratios**, the increase in perplexity is approximately equal to the sum of the contributions from QK and OV, especially for standard $A^3$-QK without RoPE, e.g., in MPT-7B, the row **both** roughly equals *qk + vo*.
> - For **larger compression ratios** (e.g., LLaMA-3.1 8B, (c_ratio = 0.4)), the contributions from QK and OV remain roughly in the same order, indicating that the local objectives provide a reliable proxy for the global perplexity impact.
>
> Table Q4a. MPT-7b, $\Delta$ ppl of compressing QK only (`qk`), VO only (`vo`), sum $\Delta ppl$ of qk and vo (`qk + vo`), $\Delta$ ppl of compressing both QK and VO (`both`).
> | Method   | 5%      | 10%     | 15%     | 20%     | Final |
> |----------|---------|---------|---------|---------|--------|
> | `qk`       | -0.004 | 0.005  | 0.040  | 0.092  | 0.40   |
> | `vo`       | 0.048  | 0.097  | 0.166  | 0.248  | 0.75   |
> | `qk + vo`  | 0.045  | 0.103  | 0.206  | 0.340  | 0.98   |
> | `both`     | 0.044  | 0.102  | 0.197  | 0.313  | 1.73   |
>
> Table Q4b. Llama-3.1-8b, $\Delta$ ppl of compressing QK only (`qk`), VO only (`vo`), sum $\Delta ppl$ of qk and vo (`qk + vo`), $\Delta$ ppl of compressing both QK and VO (`both`).
> | Method      | 5%    | 10%   | 15%   | 20%   | 40%    |
> |-------------|-------|-------|-------|-------|--------|
> | `qk`      | 0.07  | 0.16  | 0.32  | 0.56  | 13.28  |
> | `vo`     | 0.27  | 0.39  | 0.58  | 0.78  | 2.79   |
> | `qk + vo` | 0.34  | 0.55  | 0.90  | 1.34  | 16.07  |
> | `both`    | 0.35  | 0.59  | 1.00  | 1.58  | 25.07  |
>
>
> We have revised the ablation study and included a new session in **Appendix D** to reflect this point in the revised manuscript.

---

> ### Author Response · Authors · 2025-11-17
> **Q5**
>
> > How exactly is the "compression ratio" defined and matched across methods? Do you equalize total FLOPs (including attention), KV cache, and parameters, or parameter count alone?
>
> We define the compression ratio as the reduced parameter count/total parameter count. Please see reply for Q2.

---

> ### Author Response · Authors · 2025-11-17
> **Q6**
>
> > For RoPE, how robust is the CUR frequency-pair selection across datasets and sequence lengths? Any analyses showing which frequencies are retained/dropped and their impact on long-context performance?
>
> This is a very insightful question. Analyzing RoPE in the **frequency domain** has become a popular research direction. Recent works (e.g., [Round & Round We Go](https://arxiv.org/abs/2410.06205)) show that **low-frequency components** primarily capture semantic relationships, while **high-frequency components** encode positional information. These findings have inspired methods such as **partial-RoPE** ([TransMLA](https://arxiv.org/pdf/2502.07864), [MLA](https://arxiv.org/abs/2405.04434)) and strategies that **interleave RoPE and NoPE across layers** ([Rope to Nope and Back Again](https://arxiv.org/pdf/2501.18795)) to improve performance, especially for long-context scenarios.
>
> In our work, we focus on **dropping frequencies based on CUR decomposition**, which directly minimizes the loss in attention scores. Conducting a detailed frequency-based analysis in combination with CUR decomposition and exploring how to optimally merge these insights for long-context performance is a promising direction. However, it requires extensive experiments and analysis, which we leave for future work.
>
> For now, we hope that the analysis on **local attention-score objective** and **global end-to-end perplexity analysis** presented in Q4 provide useful insights into end-to-end model performance.

---

> ### Author Response · Authors · 2025-11-17
> **Q7**
>
> > Q3. We can do damping to make it invertible. In all of our experiments, they were always invertible. We can try scaling from 1 sequence and check till when the matrix is invertible.
>
> This symmetric semi-positive definite auto-correlation matrix is a well known matrix used in many recent works like GPTQ, SVD-LLM, QERA, etc. Following the [implementation in GPTQ](https://github.com/IST-DASLab/gptq/blob/2d65066eeb06a5c9ff5184d8cebdf33662c67faf/gptq.py#L98), we add damping to zero eigenvalues of the autocorrelation matrix. Additionally, we found in all of our experiments across different calibration datasets, the autocorrelation matrices were always invertible. We have added this in the **Appendix C** Approximation paragraph in the revised manuscript.

---

> ### Author Response · Authors · 2025-11-17
> **Q8**
>
> > Are you able to run more throughput experiments, e.g., with different batch sizes, sequence lengths,
> attention backends, and head dimensions (including hardware-friendly multiples)?
>
> Yes, please refer to Q3.

---

> ### Author Response · Authors · 2025-11-24
> **Reaching the End of the Public Discussion Phase**
>
> Dear Reviewer,
>
> We updated additional experiments and analysis to answer your questions and hope that our responses have sufficiently addressed the concerns you raised. We welcome more discussion if you have more questions and suggestions.
>
> As the discussion deadline is approaching, we would be very grateful if you could take a moment to review our reply. Thank you for your time and consideration.

---

> ### Author Response · Authors · 2025-11-26
> **Public Discussion Phase Ending Soon**
>
> Dear Reviewer,
>
> As the rebuttal is closing in 5 days, we would be grateful if you could kindly read our reply to your questions and advices.
>
> We addressed your concerns with additional clarifications, experiments and analysis, and highlighted the revised part in the manuscript with "@Reviewer n81d" in orange.
>
> If you have follow-up questions, we will try our best to answer them. Thank you for your time and consideration.

---

### Official Review · Reviewer_gBeN · 2025-11-01

**Soundness:** 3
**Presentation:** 4
**Contribution:** 3
**Rating:** 6
**Confidence:** 3

**Summary:**

The paper proposes a new low-rank approximation method for transformer blocks. In contrast to prior work that minimizes the approximation error of individual linear layers, this method divides the transformer blocks into functional components (attention scores, attention outputs, and MLPs) and derives analytical solutions to approximate each of them. The authors show that such an approximation produces a smaller error and can be implemented more efficiently (with less GEMM operations). It also reduces the KV cache size and might be stacked with quantization to further reduce the model size.

**Strengths:**

1. **Practical solution.** The proposed method is easy to implement, it improves the KV cache size and the throughput in tokens/sec. Furthermore, it is compatible with common architecture changes such as RoPE and grouped-query attention. The authors evaluate their method on a variety of models and benchmarks.
2. **Significance.** The paper addresses the problem of reducing the size of LLM parameters, which is crucial for deploying these models.
3. **Clarity.** The paper is well-written and describes the proposed method in a clear way.

**Weaknesses:**

1. **Compression time.** It would be valuable to extend Appendix F with comments on how compression time scales with model dimensions, as well as with the time needed to compress the Llama-3.1-70B model.

2. **Comparison to other compression methods.** While improving low-rank approximation methods might be valuable on its own, the paper doesn't directly compare $A^3$ to state-of-the-art quantization methods and doesn't clarify whether using quantization + $A^3$ produces better results than simply quantizing the model more aggressively (e.g. using less memory for storing outliers in high precision in methods like [1]).

3. **Limited novelty.** It's worth noting that the paper extends the well-known method of minimizing layer's output error and doesn't suggest fundamentally new approaches to low-rank approximation - however, publishing these results is valuable for practitioners.

4. **Typos.** L933: "Sever specs" -> "Server specs"

[1] Dettmers, Tim, et al. "GPT-3.int8(): 8-bit matrix multiplication for transformers at scale." Advances in neural information processing systems 35 (2022): 30318-30332.

**Questions:**

1. When would you recommend using $A^3$ on top of/instead of existing quantization methods?
2. How does offline compression time scale with model dimensions? How long does it take to compress the Llama-3.1-70B model?

---

> ### Author Response · Authors · 2025-11-17
> **Q1**
>
> Thank you for the careful review and enlightening questions! Here we answer them one by one. Meanwhile, the revised parts of the manuscript are colored in blue and the changes reflecting your comments are highlighted with "@Reviewer gBeN" in green. The line numbers are based on the revised version.
> > It would be valuable to extend Appendix F with comments on how compression time scales with model dimensions, as well as with the time needed to compress the Llama-3.1-70B model.
>
> Yes, to extend the compression time results in Appendix F, we additionally provide the following results to illustrate how offline compression time scales with model dimension. Using models from **1B up to 70B parameters**, we measure the time required to compress their rank to $r' = 0.8r$ on a single H100 GPU.
>
> Table W1. Compression time (in seconds) of GQA-Rope arch models on a single H100.
> | Model            | Hidden size | QK              | OV              | MLP                 | Total     |
> |------------------|-------------|------------------|------------------|----------------------|-----------|
> | Llama-3.2-1B     | 2048        | 10.41 (2.7%)     | 11.46 (3.0%)     | 358.12 (94.2%)       | 379.99    |
> | Llama-3.2-3B     | 3072        | 46.08 (6.4%)     | 52.27 (7.2%)     | 626.05 (86.4%)       | 724.40    |
> | Llama-3.1-8B     | 4096        | 100.58 (2.5%)    | 110.44 (2.8%)    | 3792.18 (94.7%)      | 4003.21   |
> | Qwen3-32B        | 5120        | 410.76 (1.0%)    | 464.27 (1.1%)    | 41381.80 (97.9%)     | 42256.83  |
> | Llama-3.1-70B    | 8192        | 1789.83 (2.4%)   | 1950.49 (2.6%)   | 71387.91 (95.0%)     | 75128.23  |
>
> The figure version is provided here: [Compression time Scaling](https://bashify.io/i/KqIsUI)
>
>
> - **Compression time grows exponentially with model size.** This is expected, since the dominant operations in the (A^3) algorithm have computational complexity **greater than $O(n^2)$**, where ($n$) denotes the model’s hidden dimension.
> - Importantly, the compression of each layer is fully independent. In practice, this allows the offline compression process to be parallelized across **n** GPUs, achieving up to n times speedup. This makes the method highly scalable in realistic multi-GPU settings, which is also how we actually ran the main experiments in Section 4.
> - For GQA architectures using RoPE, the **QK joint decomposition becomes increasingly expensive** as the hidden size (i.e., number of attention heads) grows. In large models such as the **70B variant**, QK decomposition accounts for **over 90% of the total compression time**, dominating the overall cost.
>
> These experiments and the accompanying discussion have been added to **Appendix F** in the revised manuscript.

---

> ### Author Response · Authors · 2025-11-17
> **Q2**
>
> > Comparison to other compression methods. While improving low-rank approximation methods might be valuable on its own, the paper doesn't directly compare $A^3$ to state-of-the-art quantization methods and doesn't clarify whether using quantization + $A^3$ produces better results than simply quantizing the model more aggressively.
>
> We would like to clarify that $A^3$ is a low-rank approximation method. **Low-rank approximation and quantization are generally considered as two distinct research areas**, as each possesses unique strengths and weaknesses in practical applications. Consequently, direct comparisons between the two are uncommon.
>
> Nowadays quantization works often show better accuracy–compression trade-offs in isolation ([FWSVD](https://openreview.net/forum?id=2kQOM-K9OIv), [SVD-LLM](https://proceedings.iclr.cc/paper_files/paper/2025/file/3104e1ab39875cf54fe1eb4473e7c5a1-Paper-Conference.pdf), [QTIP](https://proceedings.neurips.cc/paper_files/paper/2024/hash/6de2e84b8da47bb2eb5e2ac96c63d2b0-Abstract-Conference.html), [Quarot](https://dl.acm.org/doi/10.5555/3737916.3741096)). We would like to further discuss **the advantage of each and potential good practice of combining them**, as an extension to the existing (quantization + $A^3$) experiments in Appendix D.
>
> ### 1. Advantages of Quantization
>
> In many cases, naive low-rank decomposition methods lag behind quantization in accuracy when the goal is to **reduce memory bandwidth**. For example, **Quarot 4-bit quantization achieves a perplexity of 6.1 on LLaMA-2-7B**, which corresponds to a compression ratio of approximately **75%** for low-rank decomposition. In comparison, **SVD-LLM reaches a perplexity of 66.62 at 60% compression**, and even at 20% compression the perplexity is 7.94.
>
>
> ### 2. Advantages of Low-Rank Decomposition
> However, quantization and low-rank decomposition are **not directly comparable**, and low-rank methods bring important benefits that quantization alone cannot offer.
>
> **Hardware Independence**: Quantization typically requires specialized hardware support to realize real speedups. Without such support, dequantization introduces computational overhead—memory is saved, but actual throughput does not improve. In contrast, low-rank decomposition methods such as $A^3$ reliably offer **both memory savings and throughput speedup on any hardware**, with no need for special instructions/hardware support.
>
> **Finer Control and Extreme Compression**: For extreme compression regimes, combining low-rank methods with quantization creates a **continuous spectrum of compression levels** between discrete quantization points and can yield a way better Pareto frontier. Here we extend the “Quantization compatibility” experiments in Appendix 5. As shown in Fig. W2 sub-3-bit **quantization alone destabilizes the model**, whereas **$A^3$ + Quantization** provides a significantly better accuracy vs compression trade-off.
> - [Figure W2. Lower perplexity enabled by A3 + quantization.](https://bashify.io/i/1PEH4V)
>
> **Complementarity with Quantization**: Low-rank decomposition is not a competitor to quantization. In fact, many recent works, **[SVDQuant](https://arxiv.org/abs/2411.05007), [QERA](https://arxiv.org/abs/2410.06040), [Slim](https://arxiv.org/abs/2410.09615), [LQER](https://arxiv.org/abs/2402.02446), [OATS](https://arxiv.org/abs/2409.13652)**, among others, explore how to combine the strengths of both techniques to achieve superior overall compression. Investigating how best to integrate $A^3$ with quantization is a promising direction, but it lies outside the scope of this paper.
>
> We have appended this discussion and experiment in the Discussion section of the Quantization paragraph in the revised manuscript.

---

> ### Author Response · Authors · 2025-11-17
> **Q3-1**
>
> > Limited novelty. It's worth noting that the paper extends the well-known method of minimizing layer's output error and doesn't suggest fundamentally new approaches to low-rank approximation - however, publishing these results is valuable for practitioners.
>
> While $A^3$ is inspired by output-error minimization, it differs fundamentally in **problem formulation** and **solution, scope of optimization**, and **practical capabilities**. We highlight these differences below.
>
> ## **1. Different Problem Setup and Solution**
> ### Layer-wise output-error minimization (SVD-LLM)
>
> Methods such as SVD-LLM aim to reduce the reconstruction **error of each linear layer independently**, solving: $\min_{A,B} ||X(W - AB)||_F^2$. This objective is defined **per linear layer**, without modeling interactions across layers or across attention submodules.
>
>
> ### $A^3$: Joint optimization across multiple attention and MLP components
> In contrast, $A^3$ groups **multiple matrices within attention and MLP blocks** and optimizes them jointly. Specifically, $A^3$ minimizes the reconstruction errors of $||A - \tilde{A}||_F^2$,$||O - \tilde{O}||_F^2$,$||MLP - \widetilde{MLP}||_F^2$. This formulation yields **three key advantages**:
> - **More accurate modeling of end-to-end behavior.** Unlike SVD-LLM that minimises the output error of individual linear layers, our formulation approximates the joint behavior of multiple layers (attention and MLP components), providing a much closer proxy of model’s end-to-end performance. This more faithful modeling allows $A^3$ to significantly outperform SVD-LLM at equivalent compression ratios (parameter count).
> - **Zero-overhead solution structure.** The resulting factorization introduces **no additional runtime overhead**. $A^3$ reduces native dimension in the network. The compression ratio can **directly translate** into the same ratio in runtime speedup, throughput gain, KV-cache reduction, memory footprint saving. This is different from SVD-LLM in which rank compression is not directly equal to FLOPs and memory saving, and SVD-LLM also requires extra GEMM kernel launch as SVD-LLM splits a single linear into two smaller ones (See the GPU throughput experiments in Figure 3, Section 4) . This produces a more effective accuracy vs compression trade-off than SVD-LLM.
> - **Improved attention modeling beyond KV-cache methods.** Although $A^3$ is a whole-model compression method, its objective captures both **attention weights** and **attention outputs**. This leads to results that outperform even **recent KV-cache specific methods**, such as Clover and Palu. Here, we additionally provide KV cache compression analysis on MPT-7b and 30b across wikitext, c4 and slimpajama.

---

> ### Author Response · Authors · 2025-11-17
> **Q3-2**
>
> Table Q3: A comparison of perplexity ($\downarrow$) on WikiText2, C4, and SlimPajama on MPT-7B and MPT-30B.
> | MPT-7B |        SlimPajama        |               |               |          C4          |               |               |       Wikitext-2       |               |               |
> |--------|---------------------------|---------------|---------------|-----------------------|---------------|---------------|-------------------------|---------------|---------------|
> |CRatio | Clover                    | Palu          | $A^3$           | Clover                | Palu          | $A^3$           | Clover                  | Palu          | $A^3$           |
> | **20%** | 48.11                    | 9.67          | **8.88**          | 53.29                 | 11.74         | **10.77**         | 77.78                   | 8.73          | **8.05**          |
> | **40%** | 383                      | 11.51         | **9.90**          | 408                   | 14.18         | **12.20**         | 795                     | 10.60         | **9.19**          |
> | **60%** | 5397                     | 25.73         | **15.34**         | 4919                  | 32.26         | **18.71**         | 7895                    | 25.09         | **15.58**         |
> | **80%** | 15467                    | 5270          | **388**           | 11661                 | 3210          | **373**           | 14434                   | 13714         | **849**           |
>
>
> | MPT-30B |   SlimPajama  | |               |          C4          |               |               |       Wikitext-2       |               |               |
> |--------|---------------------------|---------------|---------------|-----------------------|---------------|---------------|-------------------------|---------------|---------------|
> |CRatio | Clover  | Palu   | $A^3$        | Clover  | Palu          | $A^3$           | Clover                  | Palu          | $A^3$           |
> | **20%** | 11.52 | 7.91   | **7.71**           | 14.53 | 9.87          | **9.59**          | 13.07                   | 7.04          | **6.73**          |
> | **40%** | 18.00 | 8.99   | **8.33**           | 22.43   | 11.30         | **10.44**         | 23.47                   | 8.40          | **7.40**          |
> | **60%** | 54.97 | 15.59 | **11.52**         | 70.65   | 18.91         | **14.22**         | 95.45                   | 18.88         | **11.28**         |
> | **80%** | 779    | 211    | **37.09**         | 732     | 253           | **42.85**         | 1524                    | 339           | **46.72**        |
>
> Here are the figure version of the table:
> - [$A^3$, Clover, Palu scale performance on MPT-7b](https://bashify.io/i/gcVE2Q)
> - [$A^3$, Clover, Palu scale performance on MPT-30b](https://bashify.io/i/0DL1yX)
>
> At the same compression ratio, Clover, Palu, and $A^3$ reduce parameters and KV-cache size by exactly the same amounts. However, their perplexity behaves very differently. Clover’s perplexity collapses immediately starting at 20% compression, while Palu performs closer to $A^3$ but not better as its objective only reduces K, V projection layer output loss.
>
> $A^3$ outperforms all three methods. Rather than minimizing error independently in each linear layer output, $A^3$ groups layers according to their interactions and functionality in the transformer block and minimizes the loss on the resulting joint outputs (Attention Score, Attention Output). It consistently achieves the lowest perplexity across all datasets and model sizes. This performance gap widens at higher compression levels and with larger model sizes: on MPT-30B, $A^3$ is the only method that stays below 50 perplexity at 80% compression, and on MPT-7B it is the only one that remains below 1k perplexity.
>
> We have added this part to the experiment session in the revised manuscript.
>
> ## **2. Practical Innovations**
> We extend $A^3$ to support **RoPE** and **GQA**, enabling compatibility with most modern transformer architectures, while recent joint low-rank methods like Clover cannot apply to RoPE/GQA. In total, $A^3$ provides **five tailored plug-and-play solutions**, allowing practitioners to directly apply it across a wide range of transformer variants.

---

> ### Author Response · Authors · 2025-11-17
> **Q4**
>
> > Typos. L933: "Sever specs" -> "Server specs"
>
> Thank you for pointing this out, we have fixed the typo in the revised manuscript.

---

> ### Author Response · Authors · 2025-11-17
> **Q5**
>
> > When would you recommend using $A^3$ on top of/instead of existing quantization methods?
>
> This is a valuable and practical question. In short, it depends on the target throughput & model performance given specific hardware platform. Quantization requires specialized hardware support to achieve acceleration but generally offer better model performance - compression trade-off; while low-rank methods like $A^3$ require no special hardware support and offer a continuous spectrum of compression levels. It is also possible to plug $A^3$ into recent works that combine quantization and low-rank methods like Slim and Qera. For details please refer to our reply in Q2.

---

> ### Author Response · Authors · 2025-11-17
> **Q6**
>
> > How does offline compression time scale with model dimensions? How long does it take to compress the Llama-3.1-70B model?
>
> Please refer to reply in Q1, in which we profiled the compression time on a single H100. In practice we parallel the compression of transformer layers across multiple H100s.

---

> ### Author Response · Authors · 2025-11-24
> **Reaching the End of the Public Discussion Phase**
>
> Dear Reviewer,
>
> We updated additional experiments and analysis to answer your questions and hope that our responses have sufficiently addressed the concerns you raised. We welcome more discussion if you have more questions and suggestions.
>
> As the discussion deadline is approaching, we would be very grateful if you could take a moment to review our reply. Thank you for your time and consideration.

---

> ### Author Response · Authors · 2025-11-26
> **Public Discussion Phase Ending Soon**
>
> Dear Reviewer,
>
> As the rebuttal is closing in 5 days, we would be grateful if you could kindly read our reply to your questions and advices.
>
> We addressed your concerns with additional clarifications, experiments and analysis, and highlighted the revised part in the manuscript with "@Reviewer gBeN" in green.
>
> If you have follow-up questions, we will try our best to answer them. Thank you for your time and consideration.

---

### Author Response · Authors · 2025-12-01
**Summary of Contributions and Rebuttals before the Technique Issue for the Area Chair**

We are aware that recent technical issues on OpenReview may have increased the burden on the Area Chair. We deeply appreciate your dedication to managing this process under such circumstances. To assist in your decision-making process, we provide a summary of our key contributions and the rebuttal before the technical issue of OpenReview.

> **1. Summary of Contributions**

We propose an analytical low-rank approximation method tailored for Transformers, $ A^3 $
-  $ A^3 $ has a three-part setup,  which formulates the problem as three separate objectives: minimizing the functional loss
of (1) QK’s attention score, (2) OV’s attention output, and (3) MLP output.
- $ A^3 $ offers closed-form solutions for the three objectives. The resultant solution reduces the hidden dimensions shared within each components: (1) QK head dim, (2) OV head dim, and (3) MLP intermediate size.
- $ A^3 $ solutions naturally enables reduced FLOPs, memory footprints, and speedup without specialised hardware, thus is a plug-and-play compression technique compatible with existing deep learning frameworks and Cuda kernels, as $ A^3 $ essentially reduces the problem size of GEMM.
- We propose variants of $ A^3 $ for diverse Transformer architectures such as RoPE and grouped query attention.
- Through extensive experiments on various LLMs, we show that $ A^3 $ outperforms SoTA low-rank methods by a significant margin.

> **2. Summary of Reviewers Concerns and Our Answers**

Here we summarize the main concerns raised by reviewers and our answers during the rebuttal

> C1. Compression ratio: Our main results focus on compression ratio $ \le $ 20% *(Reviewer voSr, 5VwV)*

- Our main results focus on compression ratio (c_ratio) $ \le $ 20% because $ A^3 $ all SoTA low-rank baselines yields broken model for c_ratio > 20%. This is the limitation of existing low-rank methods. For example, a llama-7b with c_ratio=40% yields a perplexity > 20. In this case, even $ A^3 $ still clearly outperforms the baselines, a large perplexity like 33 indicates an unusable LLM (tiny-llama 1.1b achieves perplexity around 14 with only 1/7 parameters of llama-7b)
- We additionally offer [results for c_ratio = 40%, 60%, and 80%](https://openreview.net/forum?id=no8Ooy0vaH&noteId=NCaiCSQvkF), and $ A^3 $ still outperforms SoTA low-rank methods by a clear margin.

> C2. Novelty and positioning: $ A^3 $ seems an incremental work truncating head dim, similar to CLOVER and Palu *(Reviewer gBeN, n81d, 5vWv)*

- Yes the result is truncated head dim, but different from related works minimizing linear layer's weight loss or output loss, $ A^3 $ propose minimizing functional loss of three components (attention scores, attention output, and MLP output), where each component consisting of multiple linear layers. This is a step forward from local linear layer loss to global loss, and we offer analytical solutions to these three problems.
- We offer extensive experiments to compare against these baselines and show that $ A^3 $ achieves the best model performance.

> C3: More throughput results and compression time results are needed

We offer [additional throughput results](https://openreview.net/forum?id=no8Ooy0vaH&noteId=wgZbUMXcmv) covering different GPUs, attention backend, model sizes, batch sizes, and sequence lengths.
- $ A^3 $ always enables throughput gain, unlike baselines like SVD-LLM which has extra kernel launch and under utilization problems.
- $ A^3 $ achieves more throughput gain for larger models because the reduced head dim gives extra reduced FLOPs ~$ O(\Delta d_h^2 $) in attention.

We also run [more compression time breakdown](https://openreview.net/forum?id=no8Ooy0vaH&noteId=K5AldNDSjz) to cover model size up to 70B. The QK solving is the most time consuming when the model size increases.

> C4: More ablation studies

- [More extensive comparison against Palu and CLOVER](https://openreview.net/forum?id=no8Ooy0vaH&noteId=NCaiCSQvkF)
- [Ablation study of QK and VO components](https://openreview.net/forum?id=no8Ooy0vaH&noteId=xq8GPEzbzy)

> **3. Summary of Rebuttal State before Technique Issue**

1. Reviewer gBeN and n81d did not responded to our posts and the scores remain 6 and 6.
2. Reviewer voSr responded to our posts and raised the score from 4 to 6
3. Reviewer 5VwV insisted that low-rank methods like $ A^3 $ and the SoTA baselines SVD-LLM(ICLR 2025), Palu (ICLR 2025), CLOVER (ICML 2025) should not be accepted at ICLR, and should be submitted to ACL/TMLR/EMNLP, and considered $ A^3 $ as incremental engineering work. 5VwV's score remains 2.

---

### Meta-Review · Area_Chair_Temm · 2026-01-04

**Summary:**

The paper received one “reject,” one score marginally below acceptance, and two marginally above. Reviewers raised several concerns, including the novelty of the method, its practical applicability, and the limited coverage of compression ratios in the experiments. After reviewing the authors’ responses, many of the issues identified by the reviewers remain unclear. The paper requires significant improvement before it can be considered for acceptance.

**Reviewer Concerns:**

The novelty of the method, its practical applicability, and the limited coverage of compression ratios in the experiments have not been adequately addressed.

**Reviewer Scores:**

Based on the rebuttal, it is unlikely that the reviewers will revise their scores to positive ratings.

---

### Decision · Program_Chairs · 2026-01-26

Reject